# High-level production of nervonic acid in the oleaginous yeast *Yarrowia lipolytica* by systematic metabolic engineering

Hang Su [1,2,8], Penghui Shi[1,3,8], Zhaoshuang Shen[1,3,8], Huimin Meng[1,7], Ziyue Meng[1,2], Xingfeng Han[1], Yanna Chen[4], Weiming Fan[4], Yun Fa[1,3,5], Chunyu Yang[6], Fuli Li [1,3,5 ✉] & Shi'an Wang [1,3,5 ✉]

Nervonic acid benefits the treatment of neurological diseases and the health of brain. In this study, we employed the oleaginous yeast *Yarrowia lipolytica* to overproduce nervonic acid oil by systematic metabolic engineering. First, the production of nervonic acid was dramatically improved by iterative expression of the genes ecoding β-ketoacyl-CoA synthase *Cg*KCS, fatty acid elongase gELOVL6 and desaturase MaOLE2. Second, the biosynthesis of both nervonic acid and lipids were further enhanced by expression of glycerol-3-phosphate acyltransferases and diacylglycerol acyltransferases from *Malania oleifera* in endoplasmic reticulum (ER). Third, overexpression of a newly identified ER structure regulator gene *YlINO2* led to a 39.3% increase in lipid production. Fourth, disruption of the AMP-activated S/T protein kinase gene *SNF1* increased the ratio of nervonic acid to lignoceric acid by 61.6%. Next, pilot-scale fermentation using the strain YLNA9 exhibited a lipid titer of 96.7 g/L and a nervonic acid titer of 17.3 g/L (17.9% of total fatty acids), the highest reported titer to date. Finally, a proof-of-concept purification and separation of nervonic acid were performed and the purity of it reached 98.7%. This study suggested that oleaginous yeasts are attractive hosts for the cost-efficient production of nervonic acid and possibly other very long-chain fatty acids (VLCFAs).

[1] Key Laboratory of Biofuels, Qingdao Institute of Bioenergy and Bioprocess Technology, Chinese Academy of Sciences, Qingdao 266101, China. [2] University of Chinese Academy of Sciences, Beijing 100039, China. [3] Shandong Energy Institute, Qingdao 266101, China. [4] Zhejiang Zhenyuan Biotech Co., LTD, Shaoxing 312365, China. [5] Qingdao New Energy Shandong Laboratory, Qingdao 266101, China. [6] State Key Laboratory of Microbial Technology, Institute of Microbial Technology, Shandong University, Qingdao 266237, China. [7] Present address: Qingdao Institute for Food and Drug Control, Qingdao 266073, China. [8] These authors contributed equally: Hang Su, Penghui Shi, Zhaoshuang Shen. ✉email: lifl@qibebt.ac.cn; wangsa@qibebt.ac.cn

Brain and neurological diseases affect nearly one in six of the world's population according to the American Brain Foundation (www.americanbrainfoundation.org/). Nervonic acid (cis-15-tetracosenoic acid, C24:1 Δ15), an omega-9 very long-chain monounsaturated fatty acid (VLCMFA), is rich in the white matter and myelin sheath of human brain. More and more studies show that nervonic acid benefits neurological health[1,2] and has therapeutic potential for adrenoleukodystrophy and multiple sclerosis[3,4]. Moreover, nervonic acid levels in specific tissues are associated with Alzheimer's disease[5], Parkinson's disease[6], cognition[7], mood symptoms[8,9], cardiovascular death[10] and even obesity[11].

The source of nervonic acid has attracted increasing interests due to its tremendous potentials for nutraceutical and pharmaceutical applications. Nervonic acid accounts for >40% of the total fatty acids (TFA) in seeds of the rare plant *Malania oleifera* and the herbaceous plants *Tropaeolum speciosum* and *Cardamine graeca*[12–15]. The oil extracted from *Acer truncatum* seeds containing about 5% of nervonic acid is sold as a nutraceutical in China[16]. Although a couple of plants have seeds rich in nervonic acid, supply of nervonic acid by these plants is expensive due to limitations of scale-up cultivation, climate-dependence, and harvesting cost[17]. Nervonic acid has also been identified in a few microalgae and filamentous fungi. The oleaginous microalgae *Mychonastes afer* HSO-3-1 and the filamentous fungal *Mortierella capitata* can synthesize nervonic acid accounting for 3.8% and 6.9% of the TFA, respectively[18,19]. However, the photo-autotrophic microalgae *M. afer* produces low biomass and *M. capitate* is a rare fungus lacking genetic tools, which prevent them being used as cell factories for the production of nervonic acid[20,21].

In recent years, the oleaginous yeast *Yarrowia lipolytica* has been used as a valuable host for the production of both lipid and nonlipid chemical products by metabolic engineering[22–26]. This yeast is generally regarded as safe (GRAS), grows fast, and has advantages in supply of acetyl-CoA and NADPH for anabolism[27,28]. The metabolic characteristic of *Y. lipolytica* makes it suitable for the production of fatty acids, such as poly-unsaturated eicosapentaenoic acid (EPA)[29], gamma-linolenic acid (GLA)[30] and docosahexaenoic acid (DHA)[31]. Laudably, DuPont successfully synthesized EPA (C22:6, ω-3) from C18:1-acyl-CoA by elongation and desaturation in *Y. lipolytica*, and they commercialized EPA-rich oil for food and EPA-rich *Y. lipolytica* biomass for feed applications[29,32]. In the context, *Y. lipolytica* is expected to be an appropriate host for the production of nervonic acid.

The fatty acid profiles in triacylglycerol (TAG) produced by *Y. lipolytica* mainly consist of palmitic acid (C16:0), palmitoleic acid (C16:1, ω-7), oleic acid (C18:1, ω-9), and linoleic acid (C18:2, ω-6)[33]. Several aspects should be optimized for high-production of nervonic acid by *Y. lipolytica*. First, β-ketoacyl-CoA synthases (KCS) are among the key fatty acid elongation enzymes for synthesis of VLCMFA from C18:1-acyl-CoA. The substrate preference and specificity of KCS may differ in various hosts. For example, the KCS ACJ61777.1 (GenBank No. EU871787) identified in the biennial grass *Lunaria annua* used C22:1-acyl-CoA as substrate to synthesize nervonic acid when expressed in the Ethiopia mustard *Brassica carinata*[34], while it elongated C18:1-acyl-CoA to produce mainly eicosenoic acid (C20:1, ω-9) and erucic acid (C22:1, ω-9, EA) when expressed in the yeast *Rhodosporidium toruloides*[35]. Second, most fatty acids exist in lipid forms in oleaginous microbes and therefore the selectivity of esterifying enzymes toward acyl-CoAs affects the fatty acid profiles in lipids. Third, subcellular engineering of the endoplasmic reticulum (ER) can improve the production of VLCMFA because esterification and elongation of long-chain fatty acids occur in ER[36,37].

In this study, we employed the oleaginous yeast *Y. lipolytica* as a host to de novo synthesizing nervonic acid. The metabolic pathways for overproducing nervonic acid were designed and depicted in Fig. 1. High-level production of nervonic acid was achieved by multiplex engineering strategies. This study showed the potential of oleaginous yeasts used for cost-efficient production of nervonic acid and possibly other VLCFAs.

## Experimental

**Strains and plasmids.** The *Y. lipolytica* strains po1g (Yeastern Biotech, Taibei, Taiwan, China) and po1g-G3 were used as the hosts[20]. Yeast strains constructed in this study were listed in Supplementary Table 1. Plasmids (Supplementary Table 2) were constructed using Gibson assembly method and verified by DNA sequencing. Primers used were listed in Supplementary Data1. DNA sequences (Supplementary Table 3) of heterologous genes were codon-optimized and synthesized by Beijing Genomics Institute (BGI), China.

All plasmids constructed in this study were assembled *via* Gibson assembly. The plasmid backbones and DNA fragments were amplified using the KAPA HiFi HotStart PCR kit (Kapa Biosystems, Boston, USA) and the primers described in Supplementary Data 1. The assembly was performed at 50 °C for 1 h. The assembled mixture was used to transform the *Escherichia coli* Trans1-T1 component cells (TransGen Biotech, Beijing, China) for plasmid amplification. Positive clones were verified by colony PCR and DNA sequencing.

The *Y. lipolytica* native promoter TEFin was used in controlling the expression of the *CgKCS* gene from *Cardamine graeca*[33]. The organelle target signals KDEL for ER, SKL for peroxisome, and the CoxIV signal for mitochondria were fused with *CgKCS*[26,38]. To raise the copy numbers of *CgKCS*, three *CgKCS*-expressing cassettes were assembled together, resulting in a plasmid pYL-3sKCS (Supplementary Table 2). The expression of *gELOVL6* and *MaOLE2* was driven by the inducible promoter yat1 under oleaginous conditions[29]. To overexpress *CgKCS*, *gELOVL6* and *MaOLE2*, the expression cassettes of the three genes were ligated together with backbone fragments, generating the plasmids pYL-3grDNA and pYL-3gD17 (Supplementary Table 2).

Expression of genes by homologous recombination were performed at the specific gene loci *FAD2* (fatty acid desaturase 2), *TGL4* (lipase 4), *rDNA*, *GSY1* (glycogen synthase), *SNF1* (AMP-activated S/T protein kinase), *D17* and *PEX10* (peroxisomal membrane protein) according to previous studies[29,39,40]. The plasmids pYL-rDNA-CgKCS, pYL-GSY1-CgKCS and pYL-SNF1-CgKCS were separately used to express one copy of CgKCS at the *rDNA*, *GSY1* and *SNF1* locus. The plasmids pYL-2gFAD2 and pYL-2gTGL4 were adopted to express two copies of *CgKCS* at the *FAD2* locus and the *TGL4* locus, respectively. The fusion protein of CgKCS and MaOLE2 in the plasmid pYL-D17-gMaOLE2 was expressed at the *D17* locus. To express the GPATs and DGAT2s from *M. oleifera*, the plasmids pYL-PEX10-Mo-35, pYL-PEX10-Mo-49, pYL-PEX10-Mo-88 and pYL-PEX10-Mo-90 were constructed and the gene expression cassettes were inserted at the *PEX10* locus. To express the putative regulators INO2 and INO4 of lipid synthesis, the plasmids pYL-PEX10-YlINO2 and pYL-PEX10-YlINO4 were constructed and the *PEX10* locus was employed to accommodate the gene expression cassettes.

**Culture media and conditions.** YPD medium, containing 20 g/L glucose, 20 g/L peptone, and 10 g/L yeast extract, was used in regular cultivation of *Y. lipolytica* with rotary shaking at 250 rpm and 28 °C. YNB medium used for selection of genetic transformants was made with 20 g/L glucose, 1.7 g/L yeast nitrogen

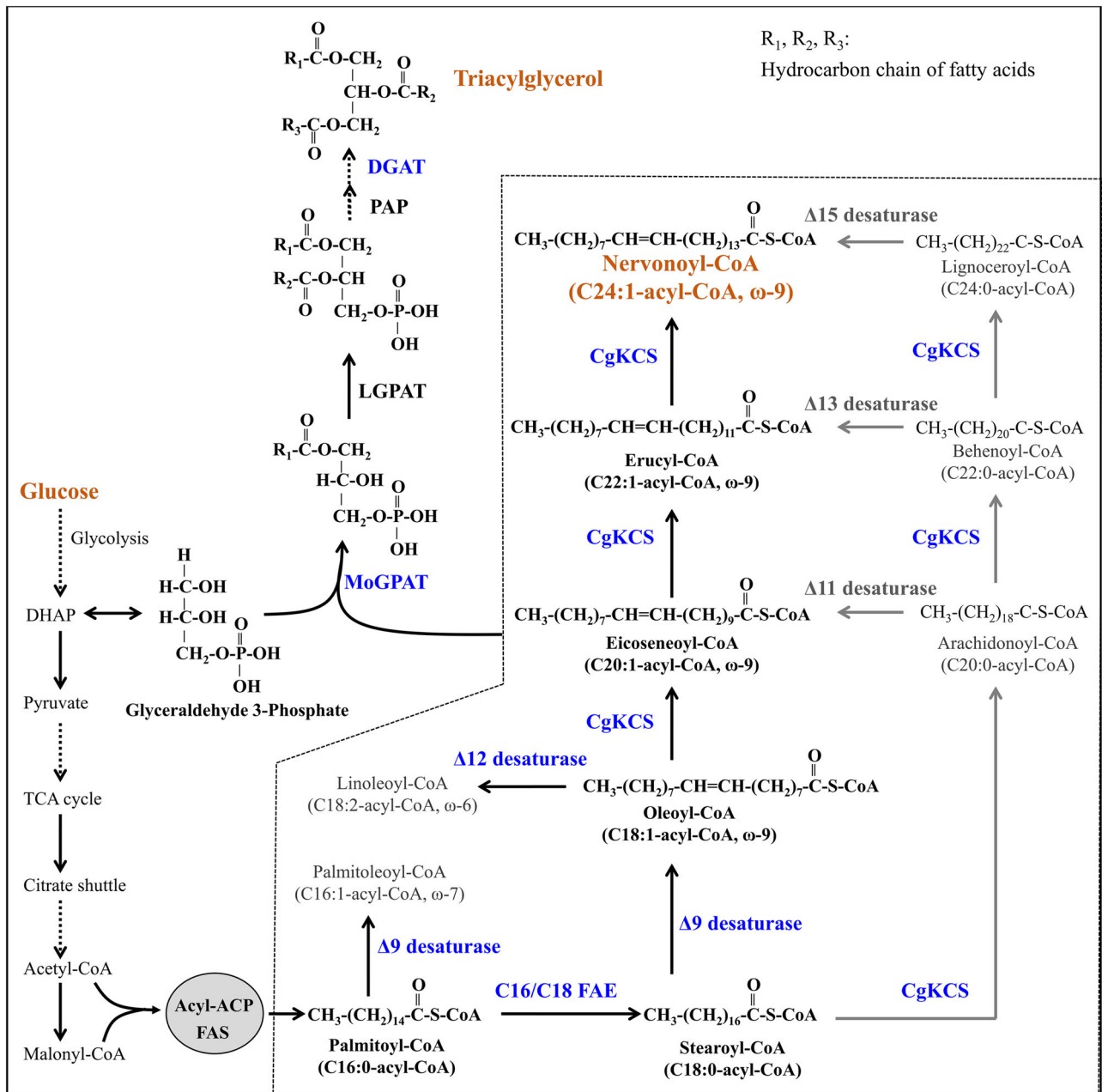

**Fig. 1 Biosynthetic pathways for nervonic acid production in engineered *Y. lipolytica*.** The solid line-box and the dotted line-box denote native pathways and heterologous pathways, respectively. The gray arrows represent competing pathways of the biosynthesis of nervonic acid. The dotted arrows show the hypothetical desaturation steps of fatty acids. FAE fatty acid elongases, KCS β-ketoacyl-CoA synthases.

(without amino acids and ammonium sulfate) (Sangon Biotech, Shanghai, China), 5 g/L ammonium sulfate, and 0.69 g/L CSM-Ura or CSM-Leu (MP Biomedicals, Solon, OH).

Shake flask cultures were carried out using the fermentative medium composed of 150 g/L glucose, 6 g/L yeast nitrogen, and 12 g/L ammonium sulfate. A single colony was inoculated into 5 mL YNB medium and cultivated at 28 °C with shaking at 250 rpm for 24 h. Afterwards, 500 μL of the seed culture was inoculated into 5 mL fresh YNB medium and cultivated at 28 °C with shaking at 250 rpm for 24 h. Then, precultures were inoculated into 30 mL of fermentative medium in 250 mL shake flasks to an optical density ($OD_{600}$) of 0.08 and incubated at 250 rpm and 28 °C for 144 h.

Bioreactor fermentation was carried out in a 50-L bioreactor (HND Bio-engineering Equipment, Jiangsu, China). The fermentative medium contained 150 g/L glucose, 6 g/L yeast extract, 12 g/L ammonium sulfate, 1.5 g/L $MgSO_4 \cdot 7H_2O$, 6 g/L $KH_2PO_4$, 3 g/L $Na_2HPO_4 \cdot 12H_2O$, 3 mg/L $CaCl_2 \cdot 2H_2O$, 1 mL/L trace metals stock (1000 ×), and 1 mL/L vitamins stock (1000 ×). One liter of trace metals stock contained 1 mg boric acid, 0.2 mg KI, 0.67 mg $FeCl_3 \cdot 6H_2O$, 0.125 mg $CuSO_4 \cdot 5H_2O$, 0.89 mg $MnSO_4 \cdot H_2O$, 0.48 mg $Na_2MoO_4 \cdot 2H_2O$, 1.42 mg $ZnSO_4 \cdot 7H_2O$, 20 mg $FeSO_4 \cdot 7H_2O$, and 0.73 mg $CoCl_2 \cdot 6H_2O$. One liter of vitamins stock contained 0.05 mg biotin, 0.8 mg calcium pantothenate, 0.004 mg folic acid, 4 mg inositol, 0.8 mg niacin, 0.4 mg *p*-aminobenzoic acid, 0.8 mg pyridoxine HCl, 0.4 mg riboflavin, 1.5 mg thiamine HCl. The vitamins stock was sterilized by 0.22 μm hydrophilic filters. The strains used in fermentation were pre-cultivated in the medium containing 20 g/L glucose, 1.7 g/L yeast nitrogen (without amino acids and ammonium

sulfate), 5 g/L ammonium sulfate, and 0.69 g/L CSM-Ura. Next, 2.3 L of the precultures were inoculated into the fermentative medium and cultivated at 28 °C. Oxygen was supplied in the form of filtered air via sparging rate of 20–80 L/min of air using agitation in 50–500 rpm range to maintain a dissolved oxygen level at 40% in 24 h and below 5% after 24 h. The pH of the cultures was constantly controlled at 5.5 using 12 mol/L NaOH. During the process of fermentation, glucose concentration was detected by an SBA-40D Biosensor (Shandong Academy of Sciences, China) every 12 h.

**Extraction and quantification of lipid and protein.** Total lipids were extracted using a previous procedure with minor modification[41]. Briefly, yeast cultures were collected by centrifugation and 0.3 g wet cells of each sample was used for lipid extraction. The wet cells were resuspended in 3 mL of 4 mol/L hydrochloric acid in glass tubes and shaken slightly for 1.5 h. Subsequently, the cells were boiled for 8 min and then were frozen at –20 °C for half an hour. Next, 6 mL of chloroform: methanol (1:1) was added to the tubes and centrifuged at 2,000 g for 10 min. The supernatant was transferred into a clean tube and 3 mL of 0.15% NaCl was added and then centrifuged. Finally, the supernatant was collected and dried using the blowing concentrator with nitrogen gas.

To detect the lipid content and composition by gas chromatography (GC), the extracted lipid was esterified by methanol to fatty acid methyl esters (FAMEs). The lipid was first incubated with 3.9 mL of methanol: sulfuric acid (98:2) at 85 °C for 3 h. Next, 1.5 mL of saturated NaCl was added and FAMEs were extracted through the addition of 1.5 mL hexane. The FAMEs were analyzed by an Agilent 7890B-GC equipped with a flame ionization detector and a capillary column HP-INNOWAX (30 m, 0.32 mm). 1 μL of sample was injected at 250 °C using helium as the carrier gas at a flow rate of 1 mL/min. The GC oven temperature was held at 140 °C for 1 min, and then ramped to 180 °C for 10 min at a speed of 10 °C/min, 210 °C for 4 min at a speed of 5 °C/min, and 250 °C for 4 min at a speed of 5 °C/min. FAMEs were identified and quantified using commercial FAME standards purchased from Sigma-Aldrich (Shanghai, China). The content of proteins was determined by the Kjeldahl method using the Kjeldahl instrument K9840 (Hanon Shandong Scientific Instruments Co., Ltd., Jinan, China).

**Structural analysis and point mutation of AtADS2.** To evaluate the substrate preference of AtADS2, the 3D structural model of AtADS2 was constructed by SWISS-MODEL server using mouse SCD (PDB ID: 4ymk) as the template[42]. Pictures of AtADS2 structures were generated with the program UCSF Chimera[43]. The docking of acyl-CoA into the binding pocket of AtADS2 was performed with the AutoDock Vina software[44]. The 3D structures of acyl-CoA and the minimized energy structures were constructed by BIOVIA Discovery Studio 4.5. Next, Auto-Dock Tools 1.5.4 was used to assign hydrogens, Gasteiger charges and rotatable bonds to acyl-CoA. A docking grid dimension was set to 64 Å × 70 Å × 64 Å and default values were used for other parameters. Based on molecular docking and characteristics of the binding pocket of AtADS2, some polar amino acids at the periphery of the binding pocket were replaced with non-polar amino acids (Supplementary Table 4). The *AtADS2* mutant genes were expressed in the *Y. lipolytica* strain YLVL6 to estimate the substrate preference according to the fatty acid profiles.

**Determination of the distribution of nervonic acid in TAG.** The distribution of nervonic acid in triacylglycerol (TAG) was detected according to a previous publication[45]. Briefly, 10 mg of

TAG and 3 mL of methanol was mixed with 10 mg of 1,3-specific lipase from *Thermomyces lanuginose* (Lipozyme TL IM; Novozymes, Bagsvaerd, Denmark) and shaken at 30 °C for 8 h. The hydrolytic products were separated by thin layer chromatography (TLC) on Partisil K6 Silica gel 60 plates (250 μm thickness, 20 × 20 cm; Merck) with developing solvent, hexane: diethyl ether: acetic acid (70:30:1, v/v). Lipids were visualized under UV light by brief exposure to iodine vapor. Then, the free fatty acids and 2-monoacylglycerols (2-MAGs) was converted to the corresponding methyl ester by incubating with 2% $H_2SO_4$-methanol solution at 85 °C for 3 h. The compositions of the fatty acid methyl esters were analyzed by GC and GC-MS.

**Illumina genome sequencing and bioinformatic analyses.** The genome sequences of representative edited clones were sequenced by Illumina technique. Standard genome sequencing and standard bioinformatic analyses were provided by Oebiotech (Shanghai, China). The filtered reads were mapped to the reference genome *Y. lipolytica* CLIB89 (W29) (GenBank: GCA_001761485.1) using the BWA 0.7.16a software. The aligned sequence reads were visualized by Integrative Genomics Viewer (IGV).

**Identification of DGAT2 and GPAT genes from *M. oleifera*.** Putative acyltransferase DGAT and GPAT have been annotated in the genome of *M. oleifera* and transcriptomic analysis showed that two *DGAT2* and two *GPAT* genes highly expressed in seeds at fast oil accumulation stage[46]. However, it is not clear whether these acyltransferases prefer esterifying C24:1-acyl-CoAs to generate nervonic acid lipids. In this study, the *DGAT2* and *GPAT* genes in *M. oleifera* were identified by comparative sequence analysis. The amino acid sequence of DGAT2 (AEE78802.1) and GPAT (AAG23437.1) was used as a BLAST query against the *M. oleifera* genome database to retrieve gene homologs, which were used as candidates for identifying C24:1-acyl-CoAs preferred acyltransferases.

**Identification of regulators of lipid synthesis in *Y. lipolytica*.** To identify regulators for lipid synthesis in *Y. lipolytica*, the regulators INO2 (NP_010408.1) and INO4 (NP_014533.1) in the phospholipid biosynthesis of *Saccharomyces cerevisiae* were used as BLAST queries against the *Y. lipolytica* W29 genome (GCA_009372015.1) to retrieve the gene homologs. Homology modeling of the retrieved YlINO2 and YlINO4 from *Y. lipolytica* was performed using SWISS-MODEL server[42]. The bHLH ScINO2-ScINO4 transcription activation complex bound the promoter region (PDB ID: 7XQ5) and the structures of TFIIIB-TBP/Brf2/DNA and SANT domain of Bdp1 (PDB ID: 5N9G) in Protein Databank (PDB) were used as templates for homology modeling of YlINO4 and YlINO2, respectively. The predicted protein structures were visualized using PyMOL Molecular Graphics System (Version 1.7.0.0). Figures 2F, 6A, 7A, B and 8A were created by Figdraw (www.figdraw.com). Structure-based multiple sequence alignment of the bHLH transcription factors was carried out on the T-COOFFEE online service[47].

**Separation and purification of nervonic acid.** Crude lipids extracted from cell sediment of the YLNA9 culture from 50 L-reactor were used to separate nervonic acid. The extracts were hydrolyzed in the solution of sodium hydroxide (pH 9.0) and ethanol at 80–85 °C with stirring for 3 h. When no stratification was observed with the naked eye, the saponification solution was acidified to pH 2.0 by sulfuric acid and heated to volatilize ethanol at 75 °C. Next, free fatty acids in the saponification solution were extracted by using *n*-hexane for three times, washed

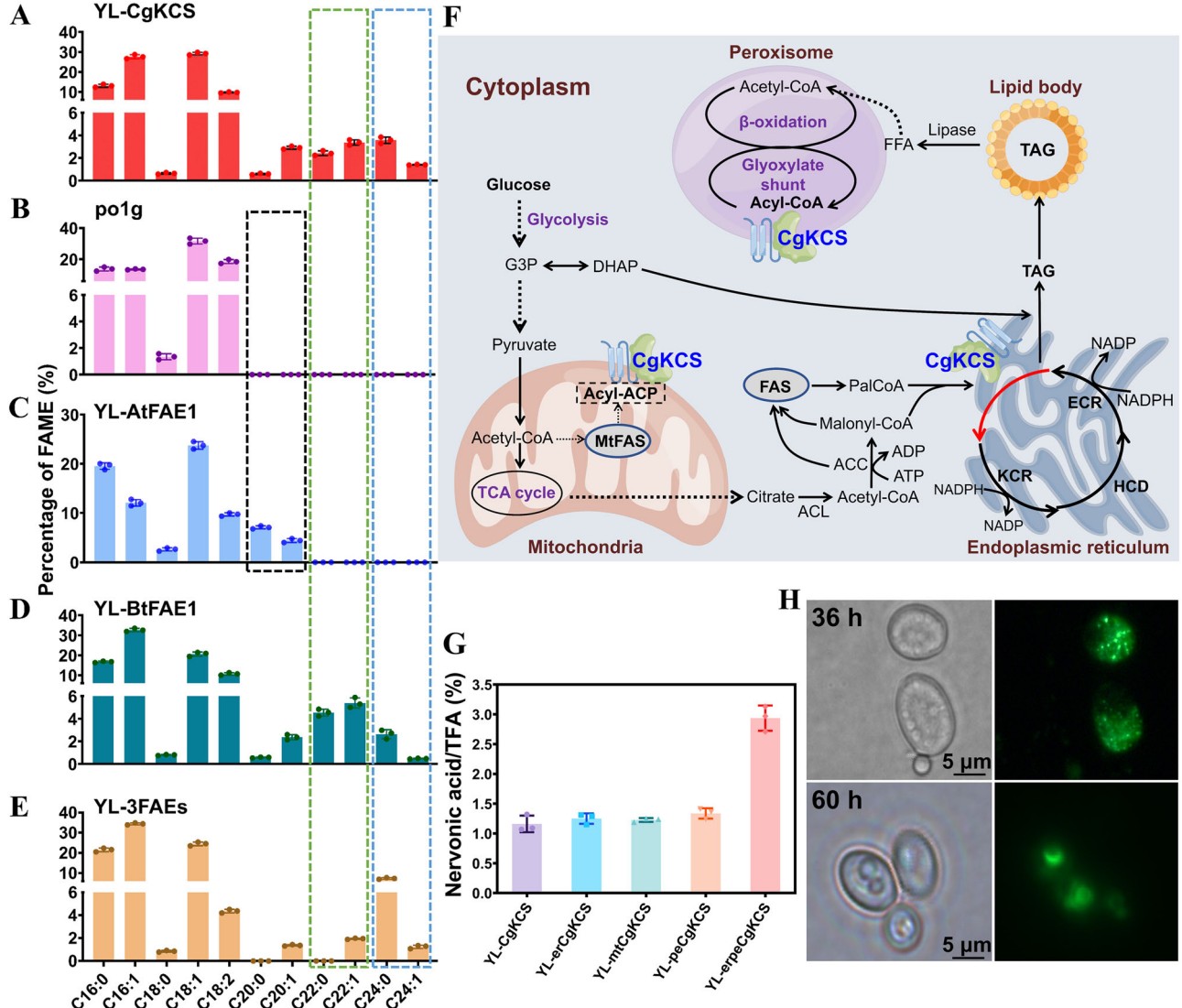

**Fig. 2 Evaluation of fatty acid elongases for production of nervonic acid. A** Expression of *CgKCS* in *Y. lipolytica* po1g. **B** The control strain po1g. **C** Expression of *AtFAE1* in po1g. **D** Expression of *BtFAE1* in po1g. **E** Co-expression of *AtFAE1*, *BtFAE1* and *CgKCS* in po1g. **F** Diagram of fatty acid metabolism related to multiple organelles and subcellular localization of *CgKCS*. ACL, ATP-citrate lyase; ACC, acetyl-CoA carboxylase; CgKCS, β-ketoacyl-CoA synthase of *C. graeca*; FAS, fatty acid synthase; FFA, free fatty acids; PalCoA, palmitoyl-CoA. **G** The relative amount of nervonic acid content in the TFA produced by the strains expressing CgKCS and CgKCS fused with organelle targeting signals. Cultivations were carried out for 144 h at 250 rpm and 28 °C in flasks. Data are mean ± s.d. from three replicates. **H** Microscopic observation of the strains expressing the fused protein CgKCS-sfGFP for evaluating the location of CgKCS in *Y. lipolytica*. CgKCS-sfGFP exhibited both punctate and accumulated localization at the early stage of cultivation (36 h) and gathered in lipid droplets at lipid accumulation stage (60 h).

and dehydrated, and then evaporated. The extracts of free fatty acids were dissolved in *n*-hexane for purification analysis.

Silica gel column chromatography was used in preliminary purification of nervonic acid. Mobile phases consisting of *n*-hexane and ammonia (0.1%, 0.5%, or 10%) or acetic acid (0.5%, 1%, or 2%) were first evaluated by TLC on silica gel plates. Next, the extracts of free fatty acids were isolated by silica gel column (26×3.0 cm i.d.) using hexane containing 1% acetic acid as the elution buffer. The initial 75 mL eluent was discarded, and following 230 mL eluent was collected and concentrated by rotary evaporator. The residues were dissolved in 1 mL of methanol for further purification by preparative chromatography. The effusion mainly containing nervonic acid was determined and collected according to the HPLC spectrogram of nervonic acid standard. Next, the collection was dried under a gentle nitrogen flow, dissolved in methanol, and filtered through 0.22-μm nylon

syringe filters for further separation by UHPLC (Ultra high performance liquid chromatography) (Thermo U3000) equipped with a UV detector through a $C_{18}$ column Poroshell 120 EC-C18 (4.6 × 150 mm, 4 μm) (Agilent Technologies, Inc.). Mobile phase consisted of acetonitrile/methanol/0.4% acetic acid/ tetrahydrofuran (80:12:5:3, v/v/v/v). Twenty microliters of samples were automatically injected. A flow rate of 0.6 mL/min of the mobile phase was used.

The reagents methanol, acetonitrile, *n*-hexane, and tetrahydrofuran used in this study are all HPLC grade and purchased from Merck Co. Acetic acid, NaOH, ammonia, sulfuric acid, and ethanol are purchased from Sinopharm Chemical Ceagent Co., Ltd. (Shanghai, China). Nervonic acid, methyl *cis*-15-tetracosenoate, and ethyl 15-tetracosenoate (>99%) were purchased from Shanghai McLean Biochemical Technology Co., Ltd., China. Silica gel was provided by Qingdao Ocean Chemical Plant, China.

**Transmission electron microscopy and fluorescence microscope**. Yeast stock cells were inoculated into 5 mL YNB medium and cultivated at 28 °C with shaking at 250 rpm for 24 h. Afterwards, 300 μL of the cultures were grown for 24 h in 30 mL of YNB medium at 28 °C and 250 rpm. Collect the cells by centrifugation and fix them by 2.5% glutaraldehyde at room temperature for 2 h. Then, gently wash the fixed cells by 0.1 M phosphate buffer solution (PBS) at pH 7.4 for three times. Further fix the cells by 1% osmic acid solution for 1 h and then wash them with PBS for three times. Dehydration of the fixed cells by acetone at the concentrations of 30%, 50%, 70%, 80%, 90%, 95% and 100%. Next, the cells were infiltrated by acetone and resin at a ratio of 7:3 (v/v) for 5 h, 3:7 (v/v) for 5 h, and then embedded by pure resin for 12 h. The specimen was stained by 2% uranyl acetate for 15 min and lead citrate for 5 min, and observed using a transmission electron microscope (FEI Tecnai G2 Spirit, OR, USA). To observe the intracellular lipid droplets, the yeast cells were stained with Nile red and observed by a fluorescence microscope (OLYMPUS BX35, Tokyo).

**Prediction and visualization of transmembrane segments in CgKCS and MaOLE2**. The transmembrane segments (TMS) of CgKCS and MaOLE2 embedded in ER were predicted using matching apparent free energy differences (ΔG prediction server v1.0) via the Sec61 translocon[48]. The anticipated TMS with ΔG ≤ 1.5 kcal/mol were considered to TMS[49]. For membrane protein topology visualization, the PROTTER online servers was used[50]. Structure homology modeling of MaOLE2 was carried out using the SWISS-MODEL online servers[42]. The three-dimensional structure of a mouse stearoyl-CoA desaturase (PBD ID: 4YMK) in PDB was used as the template for homology modeling of MaOLE2. Three-dimensional structures modeling of CgKCS was carried out using I-TASSER unified platform[51]. The predicted protein structures were visualized using the program UCSF Chimera[43] and PyMOL Molecular Graphics System (Version 1.7.0.0).

**Statistics and reproducibility**. All quantitative data are represented as the mean ± SE. Statistically significant differences between each engineered *Y. lipolytica* strains were denoted *$P < 0.05$, **$P < 0.01$ (two-tailed Student's *t*-test).

## Results

**Nervonic acid synthesis by fatty acid elongation**. The KCS, one of the fatty acid elongase (FAE) components, is considered the rate-limiting enzyme for VLCFAs synthesis[52]. We expressed the codon-optimized *CgKCS* gene from *C. graeca* in the *Y. lipolytica* strain po1g and achieved production of nervonic acid accounting for 1.4% of total fatty acids (TFA) in the strain YL-CgKCS (Fig. 2A). The strain po1g cannot synthesize eicosenoic acid (C20:1) and EA (C22:1), so it is considered that C18:1-acyl-CoA was used as the substrate of CgKCS in YL-CgKCS (Fig. 2B). The fatty acid elongases AtFAE1 in *Arabidopsis thaliana* and BtFAE1 in *Brassica tournefortii* prefer extending oleic acid to eicosenoic acid and EA, respectively[53–55]. Protein topology prediction showed that AtFAE1, BtFAE1 and CgKCS have extremely similar transmembrane structures (Supplementary Fig. 1A–E).

To test whether C20:1-acyl-CoA and C22:1-acyl-CoA can be used as substrates for KCS to produce nervonic acid, the genes *AtFAE1* and *BtFAE1* were expressed in *Y. lipolytica*. As expected, expression of *AtFAE1* and *BtFAE1* resulted in the production of eicosenoic acid (4.4% of the TFA) in YL-AtFAE1 and EA (5.4% of the TFA) in YL-BtFAE1 (Fig. 2C, D). The expression of *BtFAE1* also led to the production of a small amount of nervonic acid (0.47% of the TFA; Fig. 2D). Multisequence alignment showed

that the catalytic residues in CgKCS, AtFAE1 and BtFAE1 are extremely conservative (Supplementary Fig. 1F), while structural modeling indicated that the size of substrate-binding regions were consistent with the preference of them to different acyl-CoAs (Supplementary Fig. 2). Nevertheless, the strain YL-3FAEs co-expressing of *AtFAE1*, *BtFAE1* and *CgKCS* did not improve the production of nervonic acid (Fig. 2E), indicating that CgKCS may not be able to effectively use C20:1-acyl-CoA and C22:1-acyl-CoA as substrates in *Y. lipolytica*.

The biosynthesis of fatty acids is functionally partitioned in multiple cellular compartments, including the cytosol, mitochondria, ER, peroxisome, and lipid droplets[26,56,57]. We next investigated whether CgKCS expression in specific subcellular organelles could help direct acyl-CoA toward the formation of nervonic acid (Fig. 2F). Mitochondria was included in the subcellular localization tests because of mitochondrial FAS found in yeast and the interaction between mitochondria associated member and ER[58,59]. Expression of *CgKCS* or *CgKCS* fused with ER, peroxisomal or mitochondria signals led to similar production of nervonic acid (1.1%-1.3% of the TFA). Co-expression of *CgKCS* fused with ER and peroxisome signals resulted in an increase of nervonic acid to 2.9% of the TFA (Fig. 2G). Because expression of CgKCS without additional target signals allowed production of nervonic acid and CgKCS is a transmembrane protein (Supplementary Fig. 1A), we speculated that CgKCS may contain inherent signals for organelles localization in *Y. lipolytica*. To assess this speculation, we fused *CgKCS* to a superfolder GFP gene (*sfGFP*)[60]. CgKCS-sfGFP exhibited both punctate and accumulated localization in the early stage of cultivation and targeted in lipid droplets in the lipid accumulation stage (Fig. 2H). This observation indicates that CgKCS can target multiple cellular organelles in *Y. lipolytica*, including ER and lipid droplets. Given the effectiveness of both original CgKCS and CgKCS fusing with additional target signals, we employed combinatorial patterns to express these genes to further improve nervonic acid production.

**Evaluation of nervonic acid synthesis by desaturation**. Theoretically, nervonic acid could be synthesized from both C18:1-acyl-CoA and C18:0-acyl-CoA by expression of KCS and desaturases (Fig. 1). However, few fatty acid desaturases with VLCFA activities have been identified. To the best of our knowledge, the acyl-coenzyme A desaturase-like protein AtADS2 in *A. thaliana* and the Δ15 desaturase from *Mortierella alpina* were able to catalyze C24:0-acyl-CoA to synthesize nervonic acid[61,62]. We expressed the *AtADS2* gene in the strain YL-3FAEs due to the TFA of this strain containing 7% lignoceric acid (C24:0) (Supplementary Fig. 3A). Determination of the fatty acids in the *AtADS2*-overexpresed strains showed that nervonic acid was nearly undetected while oleic acid improved from 26% to 54% in TFA (Supplementary Fig. 3A, B), suggesting that AtADS2 desaturated C18:0-acyl-CoA by Δ9-desaturase activity. This finding is in contrast to expressing *AtADS2* in *S. cerevisiae*, in which AtADS2 presented Δ15-desaturase activity leading to nervonic acid production from C24:0-acyl-CoA[61].

Structural analysis of AtADS2 by SWISS-MODEL server showed that the three-dimensional structure of AtADS2 is highly similar to that of mouse stearoyl-CoA desaturase (mSCD, PDB ID:4ymk)[63]. Like typical SCDs, cross-sections of the AtADS2 surface have a tunnel-like substrate binding pocket, which determines the chain length of acyl-CoA substrates (Supplementary Fig. 3C). Compared with C18:0-acyl-CoA, C24:0-acyl-CoA has a longer hydrophobic hydrocarbon chain that cannot effectively combine with the binding pocket. We explored improving the substrate preference of AtADS2 toward C24:0-acyl-CoA by modifying amino acids around the pocket to

enhance its hydrophobicity (Supplementary Table 3). Unexpectedly, the content of nervonic acid decreased in all mutants, while the content of oleic acid increased from 36% to above 47% in TFA along with a slight increase of eicosenoic acid (Supplementary Fig. 3D). Since AtADS2 and the mutants tested here prefer desaturating C18:0-acyl-CoA to produce oleic acid in *Y. lipolytica*, the strategy of desaturation of C24:0-acyl-CoA for the production of nervonic acid was given up in this study.

**Modulation of endogenous pathways improved nervonic acid production.** Palmitoleic acid is an omega-7 fatty acid that cannot be used as the substrate for producing the omega-9 nervonic acid (Fig. 1). With the overexpression of *CgKCS*, palmitoleic acid increased significantly, from 13.5% to 27.3% of the TFA (Fig. 2A, B). We assumed that expression of C16/C18 FAEs that prefer to convert C16:0-acyl-CoA to C18:0-acyl-CoA together with Δ9 desaturases with high substrate preference for C18:0-acyl-CoA should decrease the synthesis of palmitoleic acid and increase the C18:1-acyl-CoA precursor for nervonic acid production (Fig. 1).

We next evaluated the substrate preference of heterologous FAEs for C16:0-acyl-CoA and fatty acid desaturases for C18:0-acyl-CoA in vivo (Supplementary Table 5). The goat ELOVL6 (gELOVL6)[64], rat ELO2 (rELO2)[65], and CpLCE1 from *Cryptosporidium parvum*[66] were chosen according to previous reports about the high elongation preference of them toward C16:0-acyl-CoA. The tested fatty acid desaturases included OLE2 from *Mortierella alpine* (MaOLE2)[67], FAT6 from *Caenorhabditis elegans* (CeFAT6)[68], and D9DMB from *Cunninghamella echinulata*[69]. All the genes were optimized according to the codon usage bias of *Y. lipolytica*.

Expression of *CeFAT6* and *rELO2* in *Y. lipolytica* po1g increased the content of linoleic acid from 21.6% in TFA to 36.4% and 41.7%, respectively. In contrast, expression of *MaOLE2* or *gELOVL6* in po1g led to an effective increase of oleic acid from 36.3% in TFA to >50% along with a decrease of palmitoleic acid from 12.1% in TFA to below 8.4% (Fig. 3A, B). Because CgKCS prefers using C18:0-acyl-CoA as the substrate for nervonic acid synthesis, *MaOLE2* and *gELOVL6* were employed in constructing nervonic acid overproducing strains. Expression of *gELOVL6* in the *CgKCS*-overexpressing strain YL-3CgK decreased the contents of palmitoleic acid and palmitic acid in TFA by 7.5% and 57.2% in the strain YL-3CgKE, respectively. Meanwhile, nervonic acid increased from 3.3% to 13.2% of the TFA in YL-3CgKE (Fig. 3C). Furthermore, expressing an additional copy of each of *gELOVL6* and *MaOLE2* in YL-3CgKE reduced palmitoleic acid by 40.5% and increased nervonic acid to 15.9% of the TFA in YL-3CgKEM, (Fig. 3C). A maximum nervonic acid titer of 0.34 g/L was achieved in flask cultures. These results confirmed the effectiveness of expression of *gELOVL6* and *MaOLE2* to decrease palmitoleic acid and improve the precursor for nervonic acid production in *Y. lipolytica*. The distribution of nervonic acid in TAG was then detected by hydrolysis using a 1,3-specific lipase as well as TLC and GC-MS analysis. The results indicated that nervonic acid produced by the engineered *Y. lipolytica* localized at sn-1 and sn-3 positions in TAG (Fig. 3D).

**Construction of nervonic acid overproduction strains by homologous recombination in lipid-overproduction strains.** Although the nervonic acid content in TFA reached 15.9% in the strain YL-3CgKEM, the lipid titer was only 2.1 g/L in flask cultures. To improve the titer of both nervonic acid and lipids, we iteratively overexpressed *CgKCS*, *gELOVL6* and *MaOLE2* in the lipid-overproduction strain po1g-G3[20] by both random integration and homologous recombination (Fig. 4A). Six copies of *CgKCS* were expressed by random genome insertion to generate

the strain YLVL6, which produced 0.69 g/L of nervonic acid (6.5% of the TFA) in flask (Fig. 4B, C). However, further expression of additional copies of *CgKCS* by random genetic integration without clearly improving nervonic acid production in YLVL9 (Supplementary Fig. 4A, B). Genome resequencing revealed that the additional *CgKCS* expression cassettes eliminated the expression of previous *CgKCS* by recombination with their TEF1 promoters (Supplementary Fig. 4C). Thus, we next adopted homologous recombination to overexpress heterologous genes at given loci. Simultaneous expression of *CgKCS*, *gELOVL6* and *MaOLE2* at the ribosomal DNA loci of YLVL6 generated the strain YLVL7, in which the content and titer of nervonic acid reached 10.0% of the TFA and 1.46 g/L in flask cultures, respectively (Fig. 4B, C). Additional expression of *CgKCS*, *gELOVL6* and *MaOLE2* at the D17 locus[40] increased nervonic acid to 11.8% of the TFA and 1.71 g/L in the strain YLVL8 (Fig. 4B, C). Next, double *CgKCS* expression cassettes were co-expressed at the fatty acid desaturase-2 (FAD2) gene locus in YLVL8, producing the strain YLVL10. Disruption of *FAD2* led to a decrease of lipid production, from 14.5 g/L in YLNA8 to 11.6 g/L in YLVL10, with a slight increase of the nervonic acid titer, from 1.71 g/L to 1.76 g/L as nervonic acid content increased to 15.2% from 11.8% of the TFA (Fig. 4B–D). The strain YLVL10 produced VLCFAs at 3.74 g/L as high as 25-folds of that in the initial strain po1g-G3 in flasks (Fig. 4E). The content of VLCFAs in YLVL10 was 32.3% of the TFA (Fig. 4F). The ratio of nervonic acid and lignoceric acid to the total VLCFAs reached 47.1% and 38.2%, respectively.

Meanwhile, we constructed a series of YLNA strains (YLNA1 to 7), in which all overexpressed genes were integrated into the genome of po1g-G3 by homologous recombination (Fig. 5A–E). In the strain YLNA7, seven copies of *CgKCS* were expressed at the genomic loci of *rDNA*, *FAD2*, *TGL4*, *GSY1* and *SNF1*. In this study, two copies of CgKCS were expressed at *FAD2*[29] gene locus, producing strain YLNA3. Strain YLNA3 had nervonic acid content up to 10.79% (per TFA), while linoleic acid content showed no significant difference between YLNA1 and YLNA3 (Fig. 5C). Afterwards, double CgKCS expression cassettes were co-expressed at the *TGL4* gene locus[29], while disruption of *TGL4* led to a slight but insignificant decrease in lipid production (Fig. 5C, D). One CgKCS expression cassette was then co-expressed at *GSY1*[29] to obtain the YLNA6 strain. Next, one copy of the CgKCS was overexpressed at *SNF1* to obtain the YLNA7 strain. The nervonic acid titer in YLNA7 was significantly higher than that in YLNA6. (Fig. 5D). A nervonic acid titer of 2.60 g/L were achieved in YLNA7 (Fig. 5A and C), which was apparently higher than the 1.76 g/L achieved in YLVL10. Interestingly, the ratio of nervonic acid to lignoceric acid (C24:0) increased from 1.38 in YLNA6 to 2.23 in YLNA7 by disruption of the AMP-activated S/T protein kinase *SNF1*[29] and the overexpression of additional copy of CgKCS by random genomic integration (Fig. 5B, C), which facilitates the purification of nervonic acid. The ratio of nervonic acid to lignoceric acid in YLNA7 was also significantly higher than that in YLVL10 (2.23 vs 1.23).

Since the fatty acid desaturase MaOLE2 prefers to desaturate C18:0-acyl-CoA to generate C18:1-acyl-CoA as the substrate of CgKCS for nervonic acid biosynthesis, it is hypothesized that the proximity of MaOLE2 and CgKCS in ER benefits the production of nervonic acid (Fig. 6A). Protein transmembrane prediction showed that MaOLE2 has five transmembrane helices (TM1–TM5) arranged in a trumpet-like shape (Fig. 6B, D). The dimetal ($Zn^{2+}$) catalytic activity sites locate in two helical turns toward the cytoplasm (Fig. 6B). CgKCS has two transmembrane segments (TM1 and TM2) at the N-terminal (Fig. 6C, E). When the transmembrane regions were removed, the truncated CgKCS

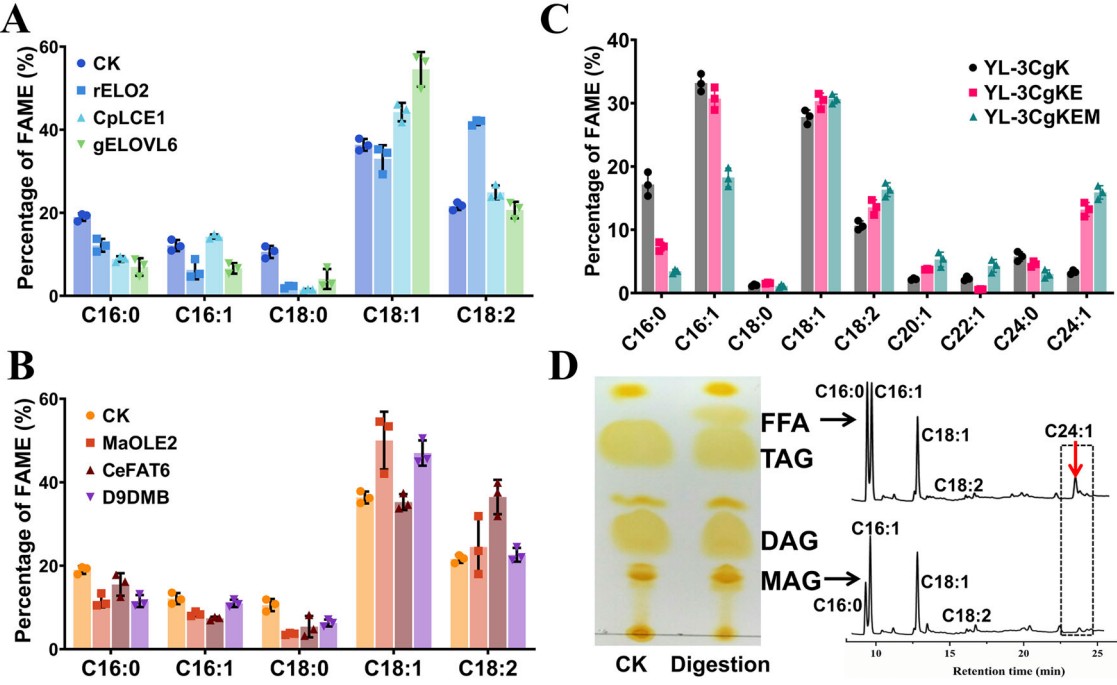

**Fig. 3 Expression of C16:0-acyl-CoA-specific FAEs and C18:0-acyl-CoA-specific fatty acid desaturases to improve nervonic acid production.**
**A** Expression of genes encoding C16:0-acyl-CoA-specific FAEs. **B** Expression of genes encoding C18:0-acyl-CoA-specific fatty acid desaturases. **C** Co-expression of *gELOVL6*, *MaOLE2*, and *CgKCS*. YL-3CgK, the *CgKCS* overexpressing strain; YL-3CgKE, expression of *gELOVL6* in YL-3CgK; YL-3CgKEM, expressing an additional copy of each of *gELOVL6* and *MaOLE2* in YL-3CgKE. **D** Determination of the distribution of nervonic acid in TAG by hydrolysis using a 1,3-specific lipase as well as TLC and GC-MS analysis. Cultivations were carried out for 144 h at 250 rpm and 28 °C in flasks. Data are mean ± s.d. from three replicates. Statistical analysis was performed by two-way ANOVA and Tukey's multiple comparison test.

maintained only a weak elongation ability (Supplementary Fig. 5). Based on the structure information, a flexibility glycine linker (GGGGGGGGGG) was used to fuse the C-terminal of MaOLE2 with the N-terminal of CgKCS. The fusion protein was expressed at the *D17* locus in YLNA7, producing the strain YLNA8. The nervonic acid content in YLNA8 (17.3%) was slightly higher than YLNA7 (16.5%) (Fig. 5C, D). The ratio of nervonic acid to lignoceric acid in YLNA8 also increased from 2.23 to 2.27 (Fig. 6F, G).

**Expansion of ER enhancing lipid and nervonic acid production.** The INO2/INO4 transcription factor complex has been demonstrated to activate phospholipid biosynthesis and ER structural alteration in *S. cerevisiae* (Fig. 7A, B)[70]. Overexpression of *INO2* and *INO4* improved the production of terpenoids, 3-hydroxypropionic acid, ethanol and lycopene by expanding the ER size or enhancing cellular stress response in *S. cerevisiae*[36,37,71]. However, the homologs of INO2/INO4 have not been identified in *Y. lipolytica*.

To recognize the ER regulatory factors in *Y. lipolytica*, a putative inositol-3-phosphate synthase YlINO1 (NC 006068.1) was first identified by two-sequence alignment. The YlINO1 was confirmed by prediction of the interaction between ScINO2/INO4 and the YlINO1 promoter by using the software JASPAR2020 (Supplementary Fig. 6A), indicating *Y. lipolytica* having INO2/INO4 complex. YlINO1 and ScINO1 (NC 001142.9) showed a sequence identity of 67.2% (Supplementary Fig. 6B). Subsequently, putative ER regulators YlINO2 (GenBank No. AOW01275.1) and YlINO4 (GenBank No. AOW06291.1) were identified by homology analysis in the genome of *Y. lipolytica*. In the GenBank database, YlINO2 was annotated as a hypothetical protein with no specific function, whereas YlINO4 was annotated as a ScINO4-like protein. YlINO4 shared a

sequence identity of 37.3% with ScINO4 and they had similar three-dimensional structure (Supplementary Fig. 7A–C). The sequence similarity between templet (Bdp1) and YlINO2 is only 15.2% (Fig. 7C). Structure-based multi-sequence alignment showed that the putative YlINO2/INO4 and the Sc INO2/INO4 shared the same essential amino acids for DNA recognition (Fig. 7D, Supplementary Fig. 7D–G), indicating that YlINO2/YlINO4 is ER transcription factors.

Next, the putative YlINO2 and YlINO4 were separately expressed at the *PEX10* locus[39] in YLNA8. Overexpression of YlINO4 did not result in clear changes in the production of lipid and nervonic acid (Fig. 7E). Overexpression of YlINO2 resulted in a 39.3% increase in lipid production from 15.2 g/L in YLNA8 to 21.1 g/L in YLNA9, as well as a significant increase in biomass from 28.0 g/L to 36.6 g/L (Fig. 7E). The rate of sugar consumption speeded up quickly from 48 h that resulted in rapid increases in the biomass and lipid titer in YLNA9 expressing YlINO2 (Fig. 7F). A lipid content of 59.5% in the DCW and a nervonic acid content of 16.5% in TFA were achieved in YLNA9 (Fig. 7G, H). The titer of nervonic acid increased by 18.2% from 2.96 g/L to 3.5 g/L in YLNA9 (Fig. 7E). The nervonic acid was confirmed by GC-MS (Supplementary Fig. 8). The ER membrane expansion was observed in the YlINO2 overexpressed strains YLNA9 by transmission electron microscopy (Supplementary Fig. 9), further indicated that YlINO2 is an ER structure regulator.

**Engineering the esterification of C24:1-acyl-CoAs.** The biosynthesis of TAG from fatty acyl-CoA and glycerol-3-phosphate requires glycerol-3-phosphate acyltransferases (GPAT), lysophospholipid acyltransferases (LPAT), phosphatidic acid phosphatase (PAP), and diacylglycerol acyltransferases (DGAT) (Fig. 8A). Most fatty acids exist in the lipid forms of TAG. Once C16-acyl-CoAs and C18-acyl-CoAs are esterified to form lipids, they

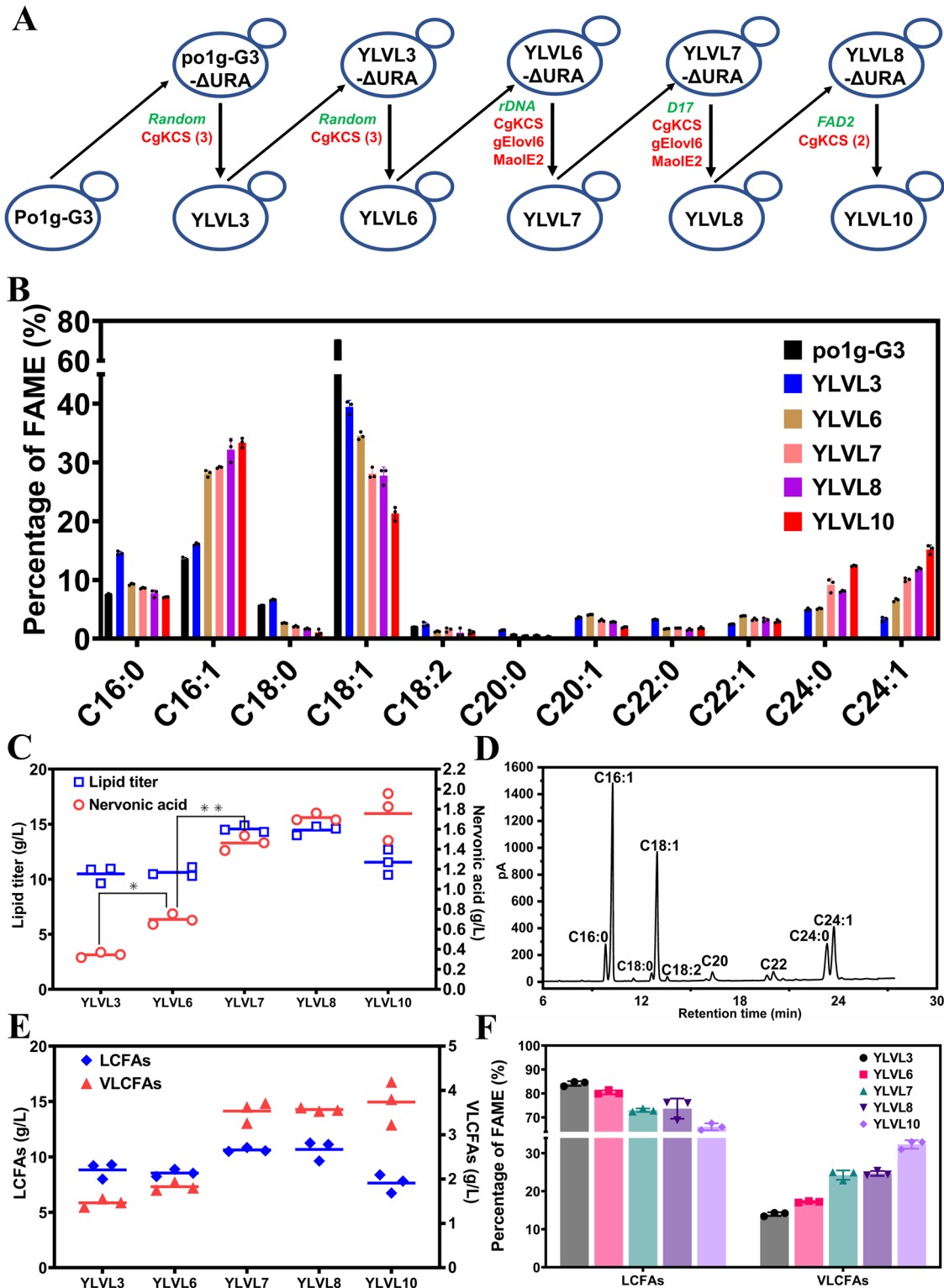

**Fig. 4 Construction of VLCFAs-producing strains expressing *CgKCS*, *gELOVL6* and *MaOLE2* by random insertion and homologous recombination.**
**A** Schematic diagram showing the construction of YLVL strains. The overexpressed genes are indicated by red symbol. The red numbers in parentheses represented the gene copies more than one on plasmids. The green italics symbol denotes genes integrated into the genomes by random integration or homologous recombination. The results of lipid and VLCFAs biosynthesis in the strains YLVL3, 6, 7, 8 and 10, including the fatty acid profiles (**B**), the titer of lipid and nervonic acid (**C**), the fatty acid composition in YLVL10 detected by gas chromatography (**D**), the titer of LCFAs and VLCFAs (**E**), and the fraction of LCFAs and VLCFAs (**F**). The yeasts were cultured for 144 h at 250 rpm and 28 °C in flasks. Data are mean ± s.d. from three replicates. Statistically significant differences between each engineered *Y. lipolytica* strains were denoted *$P < 0.05$, **$P < 0.01$ (two-tailed Student's *t*-test).

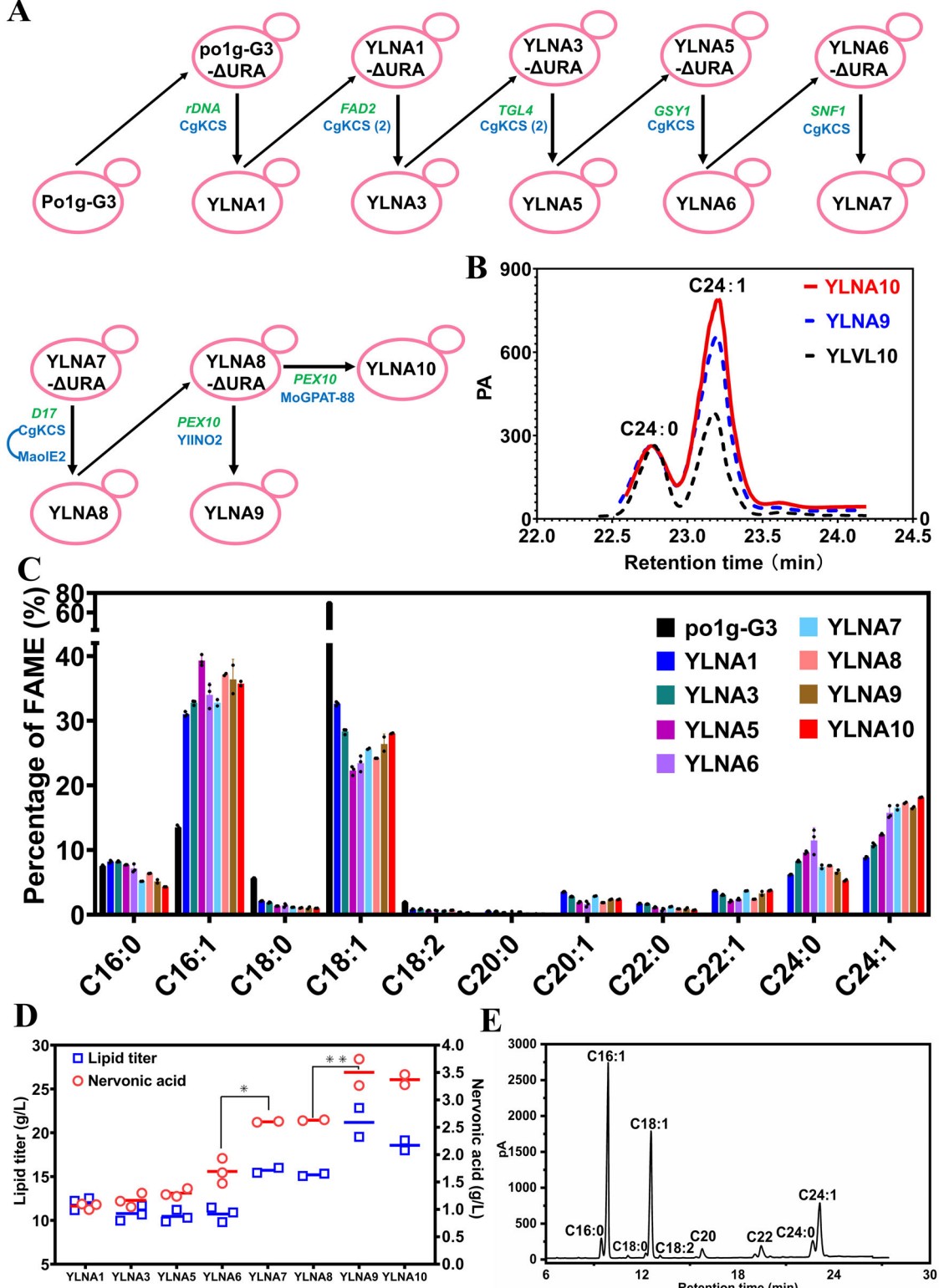

**Fig. 5 The construction of nervonic acid overproducing strains by homologous recombination. A** Schematic diagram showing the construction of YLNA strains by expressing the genes *CgKCS*, *MaOLE2*, *YlINO2* and *MoGPAT-88* at multiple genomic loci. The overexpressed genes are indicated by blue symbol. The blue numbers in parentheses represented the gene copies more than one on plasmids. The green italics symbol denotes genes integrated into the genomes by homologous recombination. **B** Comparison of the production of nervonic acid and lignoceric acid in YLNA9, YLNA10 and YLVL10. The results of lipid and VLCFAs biosynthesis in the strains YLNA1, 3, and 5 to 10, including the fatty acid profiles (**C**), the titer of lipid and nervonic acid (**D**), the fatty acid composition in YLNA10 detected by gas chromatography (**E**). The yeasts were cultured for 144 h at 250 rpm and 28 °C in flasks. For YLNA1-YLNA6 in (**C**, **D**), *n* = 3; For YLNA7-YLNA10 in (**C**, **D**), *n* = 2. Statistically significant differences between each engineered *Y. lipolytica* strains were denoted *$P < 0.05$, **$P < 0.01$ (two-tailed Student's *t*-test).

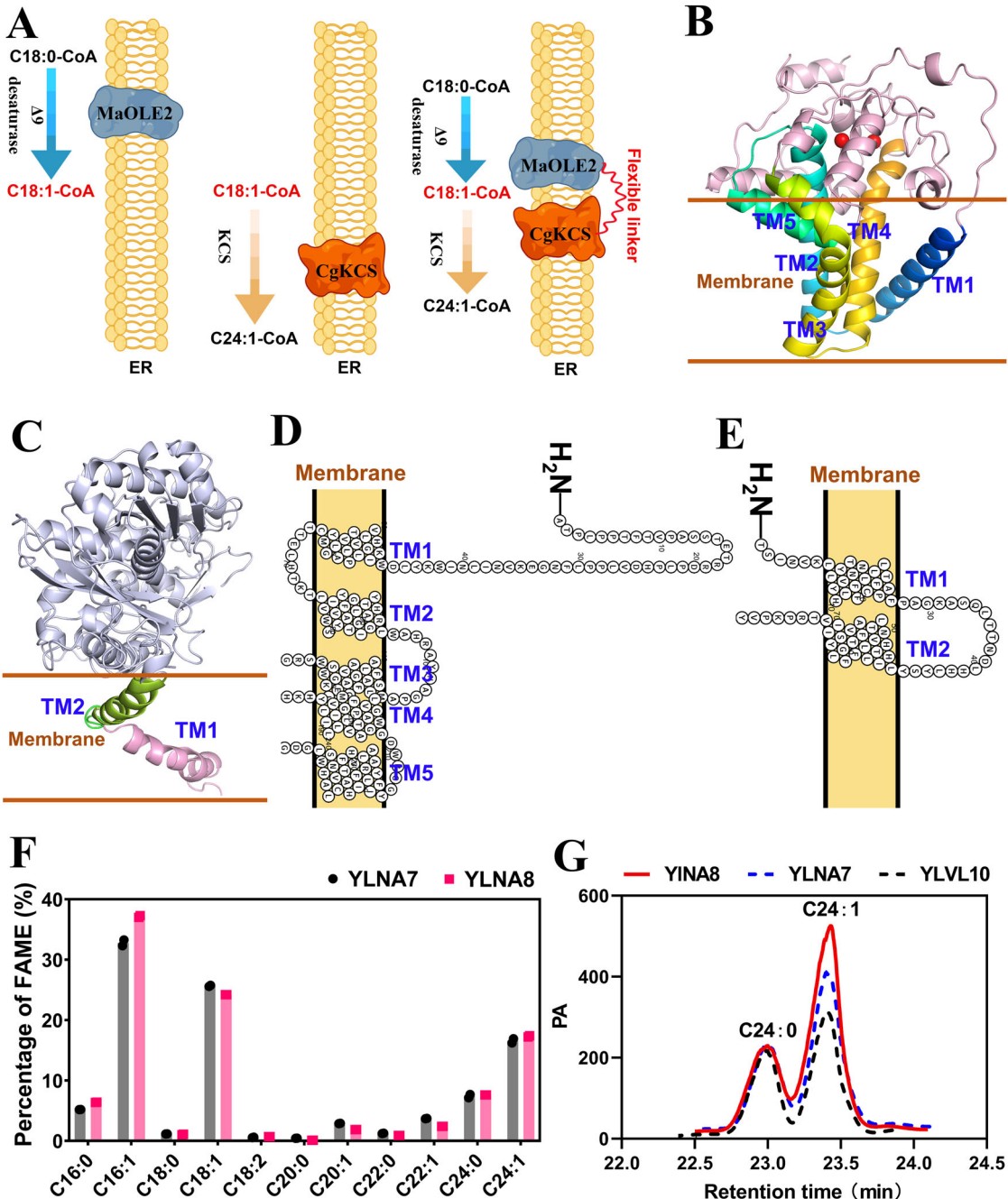

**Fig. 6 Overexpression of MaOLE2-CgKCS in ER. A** Schematic diagram showing the physical locations of MaOLE2 and CgKCS in ER. **B** Three-dimensional architecture of MaOLE2. The red spheres represent the catalytic site of zinc ions. **C** Three-dimensional architecture of CgKCS. **D** Five transmembrane domains in MaOLE2 are organized in a two-dimensional topology. **E** Two transmembrane domains in CgKCS are organized in a two-dimensional topology. **F** Comparison of the fatty acid composition between YLNA7 and YLNA8. **G** Comparison of the production of nervonic acid and lignoceric acid in YLNA7, YLNA8 and YLVL10.

cannot be elongated to C24:1-acyl-CoAs for the synthesis of nervonic acid. So, esterified enzymes with substrate preference for C24:1-acyl-CoAs rather than C16-acyl-CoAs or C18-acyl-CoAs are expected to improve the production of nervonic acid. In view of the seeds of *M. oleifera* rich in nervonic acid[12], *DGAT* and *GPAT* genes in this plant were systematically evaluated to identify C24:1-acyl-CoAs preferred ones.

Eight putative GPATs and ten putative DGAT2s sequences were extracted from the *M. oleifera* genome by homologous sequence analysis (Fig. 8B, C). Further functional prediction using UniPort in NCBI indicated that two GPATs (GenBank No.

oleifera 006088 and 006090) and two DGAT2s (GenBank No. oleifera 010035 and 015949) possibly prefer to esterify C24:1-acyl-CoAs. Both GPAT sequences 006088 and 006090 showed a sequence identity of above 55% compared with the GPAT AAG23437.1 in *A. thaliana*. The sequence identity of DGAT2 010035 and 015949 to the DGAT2 AEE78802.1 in *A. thaliana* was 25.4% and 37.8%, respectively. Separated expression of the four putative acyltransferases at the *PEX10* locus in YLNA8 increased the lipid titer by 22.1% (strain MoGPAT-88), 18.8% (strain MoGPAT-90), 15.6% (strain MoDGAT2-35) and 17.2% (strain MoDGAT2-49), respectively (Fig. 8D). A highest nervonic

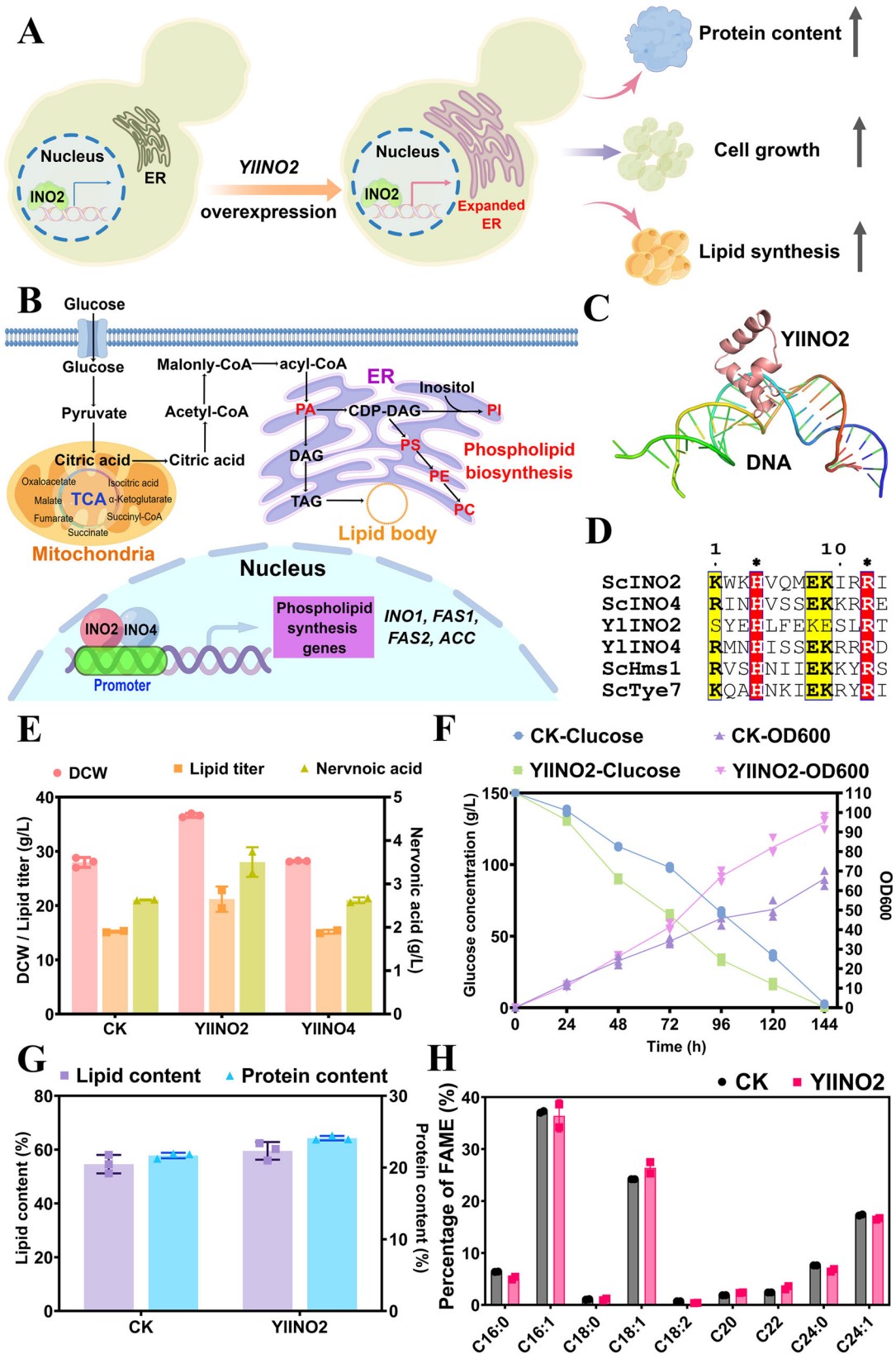

acid titer of 3.36 g/L was achieved in MoGPAT-88 (entitled YLNA10), which is 28.1% higher than that in the strain YLNA8 (Fig. 8D). The content of nervonic acid in TFA improved from 17.3% to 18.0% in YLNA10 (Fig. 8D–F). The ratio of nervonic acid to lignoceric acid in YLNA10 increased to 3.5 from 2.27 in YLNA8.

**Production of nervonic acid by two-stage fed-batch bioreactor.** The fermentative medium and conditions were optimized in flasks and a 3-L reactor. In oleaginous yeast and algae, the C/N ratio is critical for lipid accumulation[62]. In *Y. lipolytica*, nitrogen limitation (high C/N ratio media) is commonly used to induce the accumulation of intracellular lipids[72]. Growth slows down as

**Fig. 7 Identification of an ER regulator YlINO2 promoting the production of lipid. A** Schematic diagram showing the functions of INO2. **B** The regulatory network of the INO2/INO4 complex. PA, lipidphosphatidic acid; PI, phosphatidylinositol; PS, phosphatidylserine; PE, phosphatidylethanolamine; PC, phosphatidylcholine. **C** Three-dimensional architecture showing the combination of YlINO2 with DNA. **D** Structure-based multiple sequence alignment of bHLH transcription factors. Sc, *S. cerevisiae*; Yl, *Y. lipolytica*. The star symbol above the red boxes indicates the conserved residues. **E** Comparison of DCW, and the titer of lipid and nervonic acid in the CK strain YLNA8 and the strains YlINO2 (YLNA9) and YlINO4. **F** The cell growth and glucose consuming in the CK strain YLNA8 and YlINO2 (YLNA9). Data are mean ± s.d. from three replicates. **G** The content of lipid and protein in CK (YLNA8) and YlINO2 (YLNA9). **H** Comparison of fatty acid composition in CK (YLNA8) and YlINO2 (YLNA9).

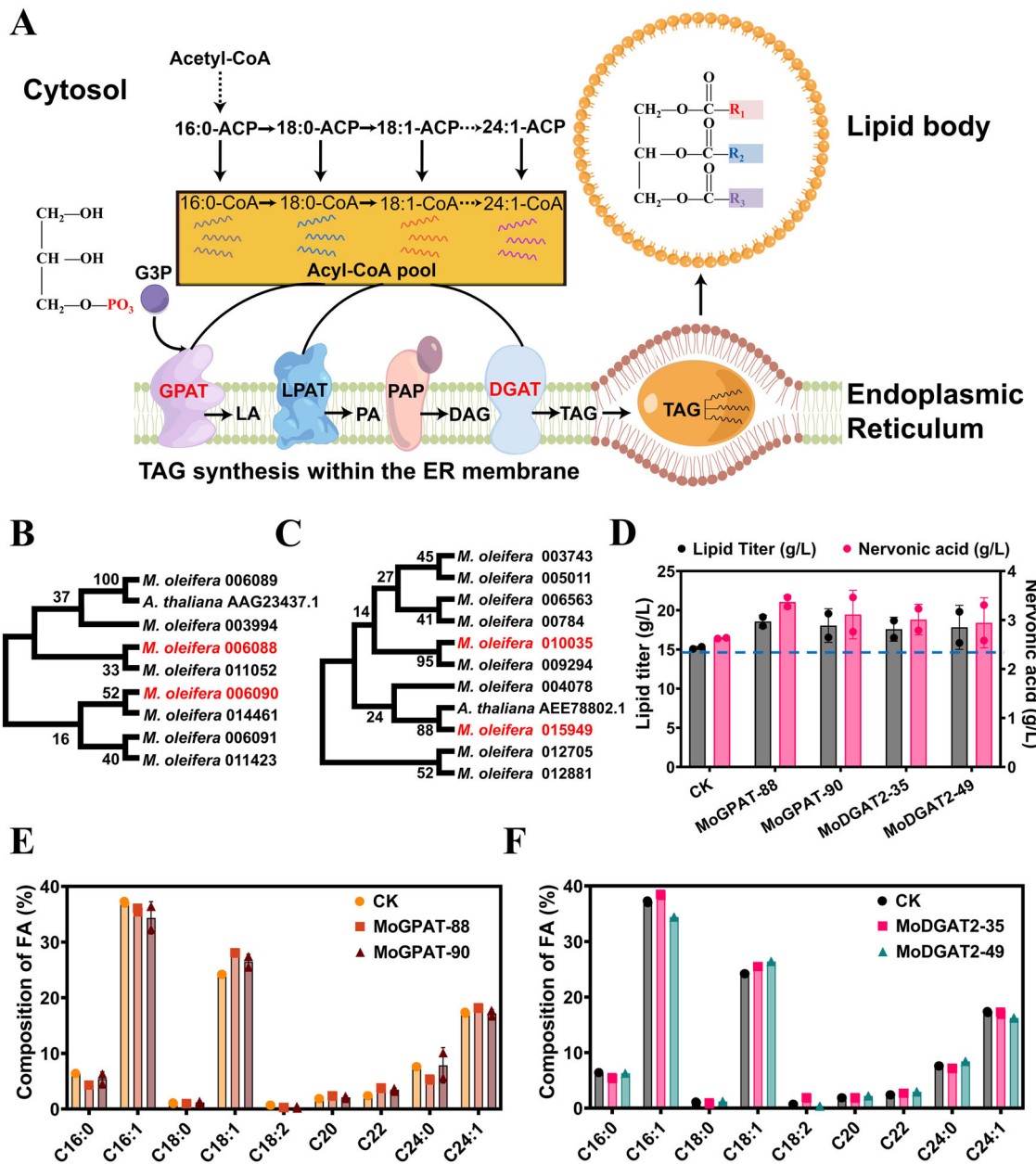

**Fig. 8 Nervonic acid and lipids improved by overexpressing GPAT and DGAT2 from *M. oleifera*. A** The biosynthesis of TAG in ER catalyzed by GPAT, LPAT, PAP and DGAT. G3P glycerol 3-phosphate, GPAT glycerol-3-phosphate acyltransferase, LPAT lysophosphlipid acyltransferase, PAP phosphatidic acid phosphatase, DGAT diacylglycerol acyltransferase. Phylogenetic trees of the acyltransferases GPAT (**B**) and DGAT2 (**C**) from *M. oleifera* constructed by the Neighbor-Joining method using the software MEGA11. The titer of lipid and nervonic acid in the acyltransferases overexpressed strains (**D**). The fatty acid composition in GPAT (**E**) and DGAT2 (**F**) overexpressed strains.

nitrogen becomes limited, and the influx of carbon is subsequently utilized mostly for the production of lipids[62]. Central composite design (CCD) and response surface methodology were used to optimize the concentrations of glucose and nitrogen sources (yeast extract and ammonium sulfate) in flasks using

strain YLVL3 (Supplementary Table 6 and 7). In two rounds of optimization experiments, the medium containing 150 g/L glucose, 6 g/L yeast extract, and 12 g/L ammonium sulfate exhibited optimal lipid production (Supplementary Fig. 10), with lipid and nervonic acid titer that were 2.9-fold and 6.7-fold the levels

obtained in the control media, respectively (Supplementary Fig. 10). The strain YLVL6 was also cultivated by fed-batch fermentation in the 3-L bioreactor and the feeding medium composition was evaluated for maximal lipid production. The results showed that feeding glucose and ammonium sulfate with a C/N molar ratio of 100:1 during fermentation resulted in an 84.6% increase in lipid titer (41.4 g/L) compared to that achieved by feeding solely glucose (Supplementary Fig. 11A). The lipid titer was further improved to 43.7 g/L by controlling the dissolved oxygen level at 20% up to 24 h and below 5% after 24 h, and the biomass reached 159.2 g/L at 120 h (Supplementary Fig. 11B). Under the optimized conditions, a nervonic acid titer of 4.2 g/L was achieved by 120 h for strain YLVL6 in the 3-L reactor.

On the basis of the optimized fermentation medium and conditions, pilot-scale fermentation in 50-L reactor was separately performed using the strains YLVL10 and YLNA9 (Fig. 9A–C). Lipid and nervonic acid synthesized constantly by YLVL10 in 192 h during the fermentation process (Fig. 9C). A VLCFAs titer of 24.0 g/L was obtained in YLVL10 at 192 h, which consisted of behenic acid (C22:0) 1.6 g/L, EA (C22:1) 2.2 g/L, lignoceric acid (C24:0) 10.0 g/L and nervonic acid (C24:1) 10.2 g/L (Fig. 9B). Lipid and nervonic acid synthesized constantly by YLNA9 in 216 h (Fig. 9C). The fastest lipid production in YLNA9 happened between 48 h and 72 h which was earlier than that of 72 h to 96 h in YLVL10, and the highest yield during this period in both YLNA9 and YLVL10 reached 0.267 g lipid /g glucose (98% of the maximum theoretical yield). A largest lipid titer of 96.7 g/L and a lipid content of 52.1% to DCW were achieved in YLNA9 (Fig. 9C, Supplementary Fig. 12). The titer of nervonic acid and VLCFAs in YLNA9 reached 17.3 g/L (17.9% of the TFA) and 28.2 g/L (29.2% of the TFA), respectively (Fig. 9C), as far as we known, both of which were the highest reported levels[35,73]. The productivity of nervonic acid is $0.135 \, \text{g} \cdot \text{L}^{-1} \cdot \text{h}^{-1}$.

**Separation and purification of nervonic acid**. Both nervonic acid and lignoceric acid have 24 carbons and the positions of them in GC spectrums are close (Fig. 5B). Separation of nervonic acid from lignoceric acid and other fatty acid components was explored. The lipids produced by the strain YLNA-9 in 50-L reactor were extracted and saponified to release free fatty acids. The standards of methyl cis-15-tetracosenoate and ethyl 15-tetracosenoate were used to estimate saponification conditions. The recoveries of methyl cis-15-tetracosenoate and ethyl 15-tetracosenoate reached 93.38 ± 6.20 % and 101.71 ± 10.53%, respectively, indicating the feasibility of the processes used in this study.

After saponification, silica gel column chromatography was used to separate nervonic acid in view of the weak polarity of it. Mobile phases consisting of n-hexane and ammonia or acetic acid were first evaluated on TLC. The results showed that using the solutions containing n-hexane and acetic acid or 0.5% ammonia successfully presented nervonic acid (Supplementary Fig. 13). Particularly, the mobile phase consisting of n-hexane and 1% acetic acid exhibited best separation efficiency and product purity and was used in silica gel column chromatography. Furthermore, discarding 75 mL of initial eluent and collecting 230 mL of subsequent eluent resulted in maximum recovery of nervonic acid. Next, the products purified by silica gel column chromatography were further separated by a C18 column using UHPLC and the eluent containing nervonic acid was collected according to the elution time of nervonic acid standard. Nervonic acid was verified by HPLC-MS/MS and the purity of the nervonic acid solution reached 98.7% (m/m) determined by HPLC (Supplementary Fig. 14). Thus, these processes successfully separated nervonic acid from lignoceric acid and other fatty acid components.

## Discussion

The number of lipid structures collected in the LIPID MAPS® Structure Database (LMSD) exceeds 47, 000 so far. Many lipids have similar structures except the chain length of fatty acids because of the non-specificity of elongation and esterification reactions toward fatty acid units. Therefore, evaluation of fatty acid elongases and esterifying enzymes is crucial for producing tailored fatty acid species[74,75].To synthesize VLCFAs, KCS is among the key enzymes used for lengthening fatty acid chains[52]. Here, we found that CgKCS prefers to directly convert C18:1-acyl-CoA to C24:1-acyl-CoA by three rounds of two-carbon addition in the yeast Y. lipolytica. Meanwhile, increase of C20:1-acyl-CoA and C22:1-acyl-CoA by expression of AtFAE1 and BtFAE1 did not help CgKCS to produce more C24:1-acyl-CoA and nervonic acid. In contrast, expression of CgKCS in R. toruloides led to approximately equal amount of EA (C22:1) and nervonic acid[35]. In addition, some KCSs prefers using C22:1-acyl-CoA as substrate to synthesize nervonic acid in plants[34]. Accordingly, we concluded that the products of KCSs depend on both enzyme specificity and the hosts, and therefore, both aspects should be seriously considered when VLCFAs are produced by metabolic engineering.

In oleaginous prokaryotes and eukaryotes, most acyl-CoAs are incorporated onto glycerol-3-phosphate (G3P) to generate TAGs as a form of energy storage. TAG biosynthesis is an evolutionarily conserved process, while acyltransferases GPAT and DGAT involved in the synthesis of TAGs do not possess rigid substrate specificity toward acyl-CoAs[76,77]. Acyltransferases with similar amino acid sequences can exhibit completely different acyl-donor specificities[78]. In the nervonic acid producing plant M. oleifera, GPAT and DGAT genes have been identified, which exhibited apparently differential transcript levels in seeds[46]. In this study, we demonstrated that expression of the GPAT Maole_006088.T1 encoding gene in Y. lipolytica effectively improved the titer and content of nervonic acid. This result indicated that regulating esterification specificity is also a strategy to overproduce tailored fatty acid species, while this strategy has not received enough attention.

Lipids play a vital function in the physiology of cells and lipid metabolism correlates with various organelles in eukaryotes[26,56,57]. The organelle ER is particularly critical in the elongation and desaturation of fatty acids and lipid formation. Biosynthesis of fatty acids require enzymes being targeted on ER. Here, we identified a hypothetical protein YlINO2 (AOW01275.1) in Y. lipolytica, which is possibly a counterpart of S. cerevisiae INO2. Overexpression of YlINO2 apparently improved the titer of both TAG and nervonic acid in Y. lipolytica, highlighting the significance of ER engineering for the production of lipids and tailored fatty acids (Table 1).

Some VLCFAs and VLCFA derived products have been utilized as lubricants, detergents, cosmetics and pharmaceuticals, such as EA, docosanol, and wax esters[79–81]. Microbial production of VLCFAs from renewable feedstocks presents a promising alternative route to sourcing these materials from plant oils or the petrochemical industry[82,83].

In this study, we engineered the oleaginous yeast Y. lipolytica to produce the VLCUFA nervonic acid and achieved the highest reported titer to date, 17.3 g/L in a 50-L bioreactor, indicating a bright prospect for commercial use. Notably, a nervonic acid titer of 13.5 g/L in 5-L bioreactor[73] was reported closely before this study posted on bioRxiv. In contrast to unsaturated fatty acids and long-chain saturated fatty acids, very long-chain saturated fatty acids (VLSFAs) have received limited attention. Interestingly, recent studies reported that arachidic acid (20: 0), behenic acid (22: 0), and lignoceric acid (24: 0) need to be distinguished from other saturated fatty acids for their potential health benefits in reducing risks of diabetes, heart disease mortality and aging[84–86]. Here, we engineered Y. lipolytica to produce the VLCFAs at 28.2 g/L in a 50-

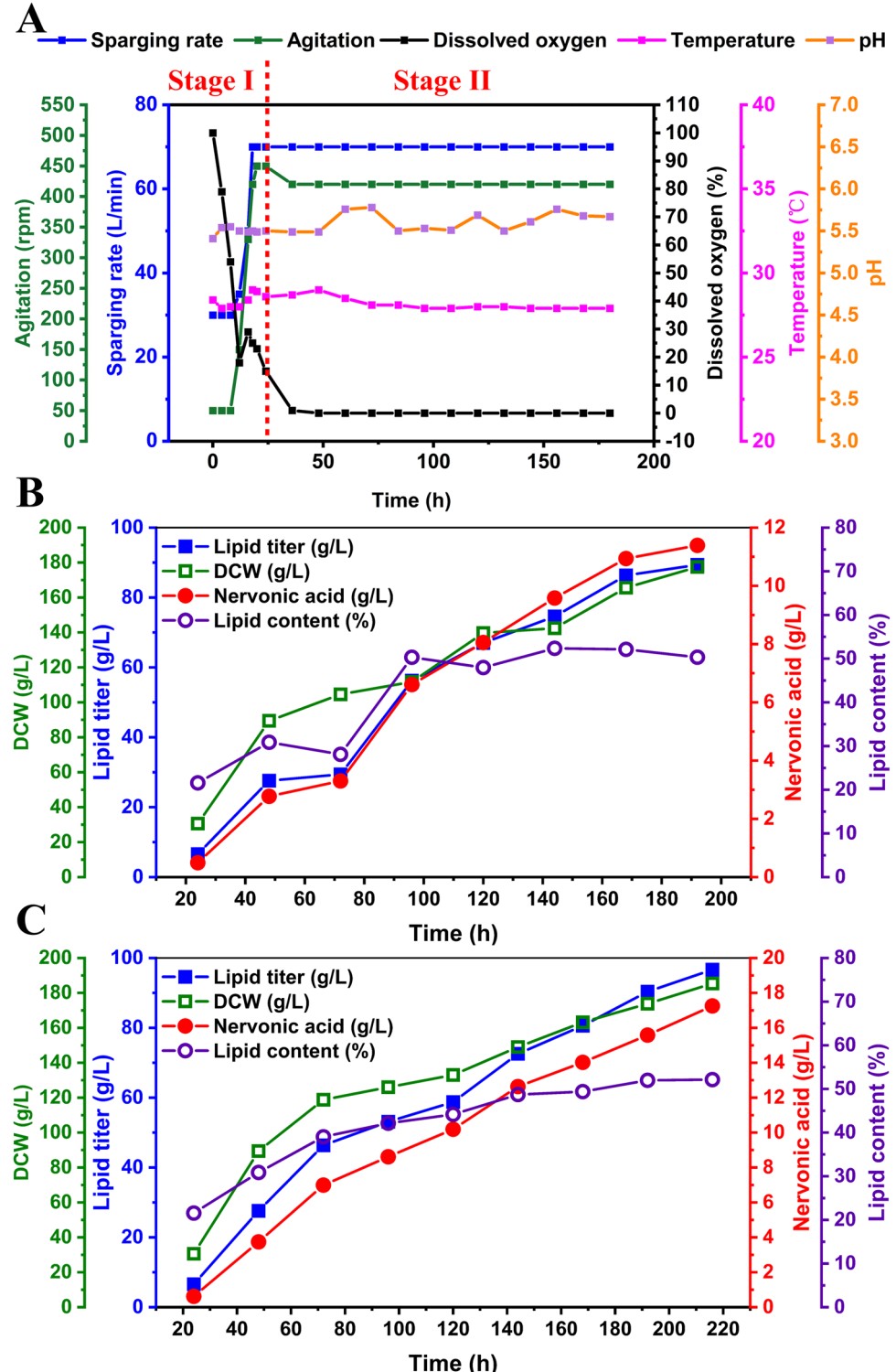

**Fig. 9 Fed-batch fermentation in a 50-L reactor. A** Conditions for controlling the two-stage fermentation process. **B** Fed-batch fermentation of YLVL10 in a 50-L bioreactor. **C** Fed-batch fermentation of YLNA9 in a 50-L bioreactor. Fermentation was conducted at 28 °C, pH 5.5. Substrate feeding to 150 g/L in every 12 h. DCW, dry cell weight.

L bioreactor, indicating this oleaginous yeast is a superior host for the production of valuable VLCFAs.

## Conclusion
In conclusion, this study engineered the oleaginous yeast *Y. lipolytica* to produce nervonic acid at a highest titer of 17.3 g/L

reported to date by multi-level metabolic engineering. Enhancing the carbon flux toward fatty acid elongation efficiently improved the production of nervonic acid and total VLCFAs. Overexpression of the newly identified ER structure regulator YlINO2 resulted in a 39.3% increase in lipid production. Proof-of-concept purification and separation of nervonic acid generated a purity of 98.7%. These results showed a bright prospect

**Table 1 Lipid production in engineered *Y. lipolytica* strains.**

| Strain | Genotype | Ratio of nervonic acid in TFA (%) | Titer of nervonic acid (g/L) | Ratio of LCFA/VLCFA |
|---|---|---|---|---|
| YLVL3 | po1g-G3, *ER-CgKCS, MT-CgKCS, PE-CgKCS* | 3.3 | 0.34 | 6.0 |
| YLVL6 | YLVL3, *ER-CgKCS, MT-CgKCS, PE-CgKCS* | 6.6 | 0.70 | 4.7 |
| YLVL7 | YLVL6, *rDNA::(CgKCS-gElovl6-MaOLE2)* | 10.0 | 1.42 | 3.0 |
| YLVL8 | YLVL7, *D17::(CgKCS-gElovl6-MaOLE2)* | 11.9 | 1.72 | 3.0 |
| YLVL10 | YLVL8, *FAD2::CgKCS*2* | 15.2 | 1.76 | 2.0 |
| YLNA1 | po1g-G3, *rDNA::CgKCS* | 8.9 | 1.06 | 3.9 |
| YLNA3 | YLNA1, *FAD2::CgKCS*2* | 10.8 | 1.16 | 3.2 |
| YLNA5 | YLNA3, *TGL4::CgKCS*2* | 12.4 | 1.30 | 2.9 |
| YLNA6 | YLNA5, *GSY1::CgKCS* | 15.7 | 1.69 | 2.2 |
| YLNA7 | YLNA6, *SNF1::CgKCS* | 16.6 | 2.60 | 2.4 |
| YLNA8 | YLNA7, *D17::(CgKCS-MaOLE2)* | 17.3 | 2.63 | 2.5 |
| YLNA9 | YLNA8, *PEX10::(YlINO2)* | 16.5 | 3.50 | 2.6 |
| YLNA10 | YLNA8, *PEX10::(MoGPAT-88)* | 18.0 | 3.37 | 2.6 |

Note: The italic formatting represents the recombination site and gene.

for the production of nervonic acid and VLCFAs by oleaginous yeasts.

**Reporting summary**. Further information on research design is available in the Nature Portfolio Reporting Summary linked to this article.

## Data availability

The authors declare that the data supporting the findings of this study are available within the Article and its Supplementary Information files. The source data for the graphs are available in Supplementary Data 2. Plasmids were deposited in Addgene (83442).

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

## Acknowledgements

This work was funded by National Key R&D Program of China (2022YFC2106200 and 2022YFE0207100), Shandong Energy Institute (SEI I202136), and Zhejiang Zhenyuan Biotech Co., LTD, China (Y86101190B).

## Author contributions

S.W., F.L. and W.F. conceived the project. S.W. and F.L. designed the experiments. H.M., H.S., X.H., P.S. and Z.M. constructed the engineered strains. H.S., C.Y., and Y.C. performed bioreactor fermentation. Z.S. and Y.F. performed the separation and purification experiments. S.W., H.S. and H.M. analyzed the data. S.W. and H.S. wrote the manuscript. S.W. and F.L. reviewed and edited the manuscript. All authors approved the final version.

## Competing interests

The authors declare competing financial interests: S.W., F.L. and W.F. have filed patent applications on this work through Qingdao Institute of Bioenergy and Bioprocess Technology, CAS and Zhejiang Zhenyuan Biotech Co., LTD, China. S.W., F.L. and W.F. have commercial interests in Zhejiang Zhenyuan Biotech Co., LTD.
