## [Peer Review File · Communications Biology]

High-level production of nervonic acid in the oleaginous yeast
Yarrowia lipolytica by systematic metabolic engineeringReviewers' comments:

Reviewer #1 (Remarks to the Author):

Nervonic acid, which benefits the treatment of neurological diseases and the health of brain, is a kind of functional fatty acid traditionally extracted from the plant. In the present study, the oleaginous yeast *Yarrowia lipolytica* was engineered for high-level production of nervonic acid by systematic metabolic engineering. The final engineered strain produced 17.3 g/L nervonic acid, the highest reported titer to date for de novo nervonic acid production. The methods are clear and the results are also well organized. For these reasons, I recommend to accept it if the following questions are addressed by the authors.

1. In Abstract section, the object after "expression of" and "disruption of" should be a gene rather than an enzyme.
2. In Abstract section, the "Cg" of the "CgKCS" should be italicized. Other similar genes or enzymes should also be changed.
3. In Abstract section, the proportion of fatty acid is as important as the titer, especially for the separation and purification. Please give the proportion of nervonic acid in total fatty acids.
4. Please provide the source of CgKCS as early as possible.
5. The full name of "KCS" and "erucic acid" should be given when it first appears, and the abbreviations should be used instead when it appears later.
6. "for production of " should be "for the production of".
7. "The fatty acid profiles in triglycerides produced by *Y. lipolytica* mainly consist of palmitic acid (C16:0), palmitoleic acid (C16:1, ω -7), and oleic acid (C18:1, ω -9)." C18:2 is also one of the most important native fatty acids in *Y. lipolytica*.
8. In figure 1, please confirm if the substrate of Δ 12 desaturase is acyl-CoA or phospholipid.
9. Please confirm whether "triacylglycerol" or "triglyceride".
10. Please confirm whether "of TFA" or "of the TFA".
11. "To the best of our knowledge, the acylcoenzyme A desaturase-like protein AtADS2 in *A. thaliana* is the only reported enzyme able to catalyze C24:0-acyl-CoA to synthesize nervonic acid." While the Δ 15 desaturase from *Mortierella alpina* has also been reported to catalyze C24:0 to C24:1, as reported in Reference 71, therefore, the above description is suggested to be adjusted.
12. Fig. 3 and 4, the fatty acid profiles of Po1g-G3 strain should be added.
13. The nervonic acid content of the strain YLNA7 should be presented.
14. Why not combine some other strategies, like co-expressing INO2 and MoGPAT-88? This may further increase nervonic acid production.
15. The T-test should be carried out. After all, some data are not significantly improved.
16. "Notably, a nervonic acid titer of 13.5 g/L in 5-L bioreactor 71 was reported closely after this study posted on bioRxiv." This description must be revised, as the Reference 71 was received on 20 March 2023, and Accepted on 1 April 2023, while the present study posted on bioRxiv was on 30 March 2023, it is after the Reference 71. Therefore, the description should be revised to be "Notably, a nervonic acid titer of 13.5 g/L in 5-L bioreactor71 was reported closely before this study posted on bioRxiv." Or just do not compare the order, I suggest.
17. The authors mentioned that they have conducted purification and separation of nervonic acid, and generated a purity of 98.7%, however, the detailed steps and procedures was not mentioned. Therefore, I suggest to delete this description to avoid of misunderstanding. Also, the authors should be clearly understand that, the nervonic acid is accumulated in the form of lipid (triacylglycerol), so the final product form is the oil product, just as the plant extract oil form product. The nervonic acid percent of the total fatty acid or the total oil is a qualitative index of the final product. Therefore, in addition to consider the final g/L nervonic acid, we should consider another important product parameter of "% of the total fatty acid or the total oil", please add this information of the final product.

Reviewer #2 (Remarks to the Author):

Hang Su and colleagues engineered the oleaginous yeast *Yarrowia lipolytica* by rational metabolic engineering to improve the production of very long-chain unsaturated fatty acids, nervonic acid. Diverse strategies, introducing key enzymes (heterologous β -ketoacyl-CoA synthase, elongate,

desaturase), enlarging a compartment for increasing lipid synthesis, and optimizing fermentation conditions successfully increased the production of nervonic acid with a high titer. This study shows several aspects which are necessary for producing specific lipids by metabolic engineering and several analyses including structural analysis of key enzymes that support the author's strategy well. The manuscript needs to be a bit more improved regarding the presentation of results depending on the strategies rather than showing all results without highlighting the conclusion of each strategy (eg. the effect of GPAT: ratio of LCFA/VLCFA, the effect of AtADS2: ratio of C18:1/C24:1).

Belows are my comments.

1. In general, it would be great if the authors can add a small table or figure (either in each figure or in another new figure) to show the effect of each engineering step (ratio of nervonic acid in TFA, titer of nervonic acid, ratio of LCFA/VLCFA as it is mentioned in discussion and conclusion).
2. The authors used specific gene loci like TGL4, GSY1, and PEX10 for homologous recombination to integrate heterologous genes. Some of the genes are highly involved in lipid metabolism in *Y. lipolytica*. So the effect of disruption of them cannot be neglected in the interpretation of the result. Please add the description of this effect and modify the data interpretation accordingly.
3. Page 2, Figure 1. The authors described the heterologous enzyme in blue in the pathway. As some native enzymes of *Y. lipolytica* are also involved in some steps in the pathway, it would be better to add the specific enzyme or gene name (eg. CgKCS, MoGPAT).
4. Page 3, The information about gene sequence is not fully described either in the main manuscript or supplementary manuscript (eg. CgKCS, AtFAE1, BtFAE1, etc). Please indicate the gene number in the database or amino acid sequence as supplementary information.
5. Page 4, Regarding the quantification of lipids and nervonic acid, the authors described "FAMES were identified and quantified using commercial FAME standards purchased from Sigma-Aldrich (Shanghai, China)". Were the FAMES in the samples calculated by the internal standard (specific chain length of FAME) or by the mixture of FAME standard mixture? Was the standard of nervonic acid used for quantifying nervonic acid in the samples, too?
6. Page 7,
 - 1) the result of expressing AtADS2 is described in Figure S3 not Figure 2E.
 - 2) please suggest the potential reason for the opposite result of expressing AtADS2 in *S. cerevisiae* and *Y. lipolytica* with relevant references. The authors discussed the assumption with structural analysis of AtADS2 and mSCD which is not verified by experiment in this study.
 - 3) The authors investigated the change of AtADS2's conformation by site-specific mutation. How were these mutation sites decided?
7. Page 8, Figure 3 (c), please described the strain's abbreviation in the legend for better understanding of readers.
8. Page 10. "Interestingly, the ratio of nervonic acid to lignoceric acid (C24:0) increased from 1.38 in YLNA6 to 2.23 in YLNA7 by disruption of the AMP-activated S/T protein kinase SNF1 (Fig. 5C and 5D),"YLNA7 was constructed by disrupting SNF1 and the overexpression of additional copy of CgKCS, therefore, the positive effect on nervonic acid production cannot be simply explained by SNF disruption. The comparison between YLNA7 and YLVL10 looks more reasonable, but still needs to be described the difference of random integration or homologous recombination during the strain construction.
9. Page 11. Figure 5 (A), please indicate the fusion protein differently (eg. CgKCS-MaoIE2).
10. Page 13. Production of nervonic acid by two-stage fed-batch bioreactor. The authors optimized the media, mostly the C/N ratio for higher lipid accumulation. Please described why the C/N ratio is affecting the lipid accumulation in *Y. lipolytica* and how it was used for lipid production with relevant references.
11. Table S6 & S7, please add the C/N ratio in each condition.
12. Fig S8, Please compare the mass spectra of nervonic acid in standard and the samples.
13. Fig S9, for supporting the enlargement of ER after INO2 overexpression, can you please provide the approximate lengths with the standard deviation of ER from multiple images? Or multiple images from the same strain can show the enlargement of ER overall?

Minor typing error

Page 13. "The titer of nervonic acid increased 18.2% from 2.96 g/L to 3.5 g/L in YLNA9 (Fig. 7E)."

-> increased by 18.2%

Page 17. Conclusion, "Enhancing the carbon flux toward fatty acid elongation efficiently..."

-> toward

Response to reviewers

Reviewer: 1

Comments to the Author

Nervonic acid, which benefits the treatment of neurological diseases and the health of brain, is a kind of functional fatty acid traditionally extracted from the plant. In the present study, the oleaginous yeast *Yarrowia lipolytica* was engineered for high-level production of nervonic acid by systematic metabolic engineering. The final engineered strain produced 17.3 g/L nervonic acid, the highest reported titer to date for de novo nervonic acid production. The methods are clear and the results are also well organized. For these reasons, I recommend to accept it if the following questions are addressed by the authors.

Comment 1: In Abstract section, the object after "expression of" and "disruption of" should be a gene rather than an enzyme.

Response: We changed the description to “expression of the genes encoding.....” and “disruption of the AMP-activated S/T protein kinase gene *SNFI*”. (please see Page 1, Abstract).

Comment 2: In Abstract section, the "Cg" of the "CgKCS" should be italicized. Other similar genes or enzymes should also be changed.

Response: The "Cg" of the "CgKCS" was italicized throughout the manuscript.

Comment 3: In Abstract section, the proportion of fatty acid is as important as the titer, especially for the separation and purification. Please give the proportion of nervonic acid in total fatty acids.

Response: The proportion of nervonic acid in total fatty acids (17.9% of TFA) was added (please see Page 1, Abstract).

Comment 4: Please provide the source of CgKCS as early as possible.

Response: The source *Cardamine graeca* of CgKCS was shown in “strains and plasmids” section of the Experimental chapter (Page 3).

Comment 5: The full name of "KCS" and "erucic acid" should be given when it first appears, and the abbreviations should be used instead when it appears later.

Response: According to the suggestion, the full name β -ketoacyl-CoA synthase of "KCS" and the abbreviation EA of "erucic acid" were used (Page 3). The corresponding abbreviations were used when they appear later (Page 5, 12, 15, and 16).

Comment 6: “for production of” should be “for the production of”.

Response: The expression was corrected (Page 1, Page 2, Page 7, Page 17, Page 18).

Comment 7: “The fatty acid profiles in triglycerides produced by *Y. lipolytica* mainly consist of palmitic acid (C16:0), palmitoleic acid (C16:1, ω -7), and oleic acid (C18:1, ω -9).” C18:2 is also one of the most important native fatty acids in *Y. lipolytica*.

Response: Linoleic acid (C18:2, ω -6) was added (Page 2).

Comment 8: In figure 1, please confirm if the substrate of Δ 12 desaturase is acyl-CoA or phospholipid.

Response: The substrate of Δ 12 desaturase is acyl-CoA (Refs: The structural basis of fatty acid elongation by the ELOVL elongases. *Nat Struct Mol Biol.* 2021, 28:512-520; Structural and mechanistic insights into the biosynthesis of unsaturated fatty acids. *IUBMB Life.* 2022, 74:1036-1051.).

Comment 9: Please confirm whether "triacylglycerol" or "triglyceride".

Response: The main difference between triacylglycerol and triglyceride is that triacylglycerol is the correct chemical name for an ester derived from glycerol bound to three fatty acids whereas triglyceride is the common name for the substance.

Triglycerides are the main constituent of the animal and vegetable fats in the diet. So, it is accurate to use triacylglycerol. We replaced "triglyceride" with "triacylglycerol" (Page 2).

Comment 10: Please confirm whether "of TFA" or "of the TFA".

Response: We used “of the TFA” instead of "of TFA". (Page 1, Page 13).

Comment 11: “To the best of our knowledge, the acylcoenzyme A desaturase-like protein AtADS2 in *A. thaliana* is the only reported enzyme able to catalyze C24:0-acyl-CoA to synthesize nervonic acid.” While the $\Delta 15$ desaturase from *Mortierella alpina* has also been reported to catalyze C24:0 to C24:1, as reported in Reference 71, therefore, the above description is suggested to be adjusted.

Response: The expression was revised to “To the best of our knowledge, the acylcoenzyme A desaturase-like protein AtADS2 in *A. thaliana* and the $\Delta 15$ desaturase from *Mortierella alpina* were able to catalyze C24:0-acyl-CoA to synthesize nervonic acid (Page 7)”.

Comment 12: Fig. 4 and 5, the fatty acid profiles of Po1g-G3 strain should be added.

Response: The fatty acid profiles of Po1g-G3 strain were added in Fig. 4 and 5 in the revised manuscript.

Comment 13: The nervonic acid content of the strain YLNA7 should be presented.

Response: The nervonic acid content of the strain YLNA7 was presented in Fig.5C (Wathet blue column) and this result was described correspondingly in the revised manuscript. Please see the page 9 “The nervonic acid content in YLNA8 (17.3%) was slightly higher than YLNA7 (16.5%) (Fig. 5C and 5D).”

Comment 14: Why not combine some other strategies, like co-expressing INO2 and MoGPAT-88? This may further increase nervonic acid production.

Response: Thanks for this valuable suggestion and we will test it.

Comment 15: The T-test should be carried out. After all, some data are not significantly improved.

Response: The T-tests of data used in Fig.4 and Fig.5 were carried out (Page8 and Page10).

Comment 16: “Notably, a nervonic acid titer of 13.5 g/L in 5-L bioreactor⁷¹ was reported closely after this study posted on bioRxiv.” This description must be revised, as the Reference 71 was received on 20 March 2023, and Accepted on 1 April 2023, while the present study posted on bioRxiv was on 30 March 2023, it is after the Reference 71. Therefore, the description should be revised to be “Notably, a nervonic acid titer of 13.5 g/L in 5-L bioreactor⁷¹ was reported closely before this study posted on bioRxiv.” Or just do not compare the order, I suggest.

Response: We accepted this suggestion and using the following description “Notably, a nervonic acid titer of 13.5 g/L in 5-L bioreactor⁷¹ was reported closely before this study posted on bioRxiv.”

Comment 17: The authors mentioned that they have conducted purification and separation of nervonic acid, and generated a purity of 98.7%, however, the detailed steps and procedures was not mentioned. Therefore, I suggest to delete this description to avoid of misunderstanding. Also, the authors should clearly understand that the nervonic acid is accumulated in the form of lipid (triacylglycerol), so the final product form is the oil product, just as the plant extract oil form product. The nervonic acid percent of the total fatty acid or the total oil is a qualitative index of the final product. Therefore, in addition to consider the final g/L nervonic acid, we should consider another important product parameter of “% of the total fatty acid or the total oil”, please add this information of the final product.

Response: The aim of analyzing nervonic acid in free fatty acid form in this study was

to verify that nervonic acid can be separated from lignoceric acid (C24:0). The detailed methods and procedures were described in Experimental section. (From page 4 to 5). Additionally, the physiological and pharmacological functions of fatty acids are not necessarily in the form of TAG. Free fatty acids or ethyl ester forms of fatty acids are also important, such as docosahexaenoic acid ethyl ester (DHA-EE) and ethyl ester of nervonic acid. Several publications involving the functions of nervonic acid in free fatty acid or ethyl ester were summarized as follows:

- a) Nervonic acid ameliorates motor disorder with Parkinson's disease. In this study, free fatty acid NA (90%) was purchased from Hengke biotechnology Co. (Shanghai, China, CAS. no: 506-37-6). (Hu, D., Cui, Y. & Zhang, J. Nervonic acid ameliorates motor disorder in mice with Parkinson's disease. *Neurochem. J.* 2021, 15, 317-324)
- b) The cell viability was significantly increased in NA-treated cells, suggesting NA had a potential activity to upregulate the ROS elimination pathway to against oxidative stress. In this study, free fatty acid NA (>99%) was obtained from Nu-Chek Prep (MN, USA). (Umemoto H, Yasugi S, Tsuda S, Yoda M, Ishiguro T, Kaba N, Itoh T. Protective effect of nervonic acid against 6-hydroxydopamine-induced oxidative stress in PC-12 cells. *J Oleo Sci.* 2021, 70:95-102)
- c) Nervonic acid limits weight gain in a mouse model of diet-induced obesity. Increasing dietary NA may be an effective way to improve the management of obesity and associated metabolic complications including diabetes. In this study, ethyl ester of nervonic acid (purity greater than 99%+) was obtained from Nu-Chek Prep (Elysian, MN). (Keppley LJW, Walker SJ, Gademsey AN, Smith JP, Keller SR, Kester M, Fox TE. Nervonic acid limits weight gain in a mouse model of diet-induced obesity. *FASEB J.* 2020, 34:15314-15326)
- d) NA can also function as a non-competitive inhibitor of human immunodeficiency virus type-1 reverse transcriptase (HIV-1 RT) in a dose-dependent manner. In this study, nervonic acid (NA, analytical grade) were purchased from Wako (Kasai, N., Mizushina, Y., Sugawara, F., and Sakaguchi, K. Three-dimensional structural model analysis of the binding site of an inhibitor, nervonic acid, of both DNA

- polymerase beta and HIV-1 reverse transcriptase. *J. Biochem.* 2022, 132, 819-828)
- e) NA can significantly ameliorate LPS-induced neuroinflammation and deterioration of learning and memory, and exerts a neuroprotective function through regulation of multiple gene transcription and metabolism pathways. In this study, NA ($\geq 99.0\%$ (GC)) were purchased from Sigma-Aldrich Co., St. Louis, MO, United States. (Wang X, Li Z, Li X, Liu X, Ying M, Cao F, Zhu X, Zhang J. Integrated metabolomics and transcriptomics reveal the neuroprotective effect of nervonic acid on LPS-induced AD model mice. *Biochem Pharmacol.* 2023, 209:115411.)

“The titer of nervonic acid and VLCFAs in YLNA9 reached 17.3 g/L (17.9% of the TFA) and 28.2 g/L (29.2% of the TFA), respectively (Fig. 9C), as far as we known, both of which were the highest reported levels.”. (page 12).

Reviewer: 2

Comments to the Author

Hang Su and colleagues engineered the oleaginous yeast *Yarrowia lipolytica* by rational metabolic engineering to improve the production of very long-chain unsaturated fatty acids, nervonic acid. Diverse strategies, introducing key enzymes (heterologous β -ketoacyl-CoA synthase, elongase, desaturase), enlarging a compartment for increasing lipid synthesis, and optimizing fermentation conditions successfully increased the production of nervonic acid with a high titer. This study shows several aspects which are necessary for producing specific lipids by metabolic engineering and several analyses including structural analysis of key enzymes that support the author's strategy well. The manuscript needs to be a bit more improved regarding the presentation of results depending on the strategies rather than showing all results without highlighting the conclusion of each strategy (eg. the effect of GPAT: ratio of LCFA/VLCFA, the effect of AtADS2: ratio of C18:1/C24:1).

Comment 1: In general, it would be great if the authors can add a small table or figure (either in each figure or in another new figure) to show the effect of each engineering step (ratio of nervonic acid in the TFA, titer of nervonic acid, ratio of LCFA/VLCFA as it is mentioned in discussion and conclusion).

Response: According to the suggestion, a summary table was added in the revised manuscript (Table 1, page 16).

Comment 2: The authors used specific gene loci like TGL4, GSY1, and PEX10 for homologous recombination to integrate heterologous genes. Some of the genes are highly involved in lipid metabolism in *Y. lipolytica*. So, the effect of disruption of them cannot be neglected in the interpretation of the result. Please add the description of this effect and modify the data interpretation accordingly.

Response: We have added the description the effect of disruption of specific gene loci in the revised manuscript (page 9).

Comment 3: Page 2, Figure 1. The authors described the heterologous enzyme in blue in the pathway. As some native enzymes of *Y. lipolytica* are also involved in some steps in the pathway, it would be better to add the specific enzyme or gene name (eg. CgKCS, MoGPAT).

Response: The specific enzymes or gene names were added according to the suggestion (Figure 1, page 2)

Comment 4: Page 3, The information about gene sequence is not fully described either in the main manuscript or supplementary manuscript (eg. CgKCS, AtFAE1, BtFAE1, etc). Please indicate the gene number in the database or amino acid sequence as supplementary information.

Response: We have added the gene sequences in supplementary data (Table. S8, Supplementary data Page 15-22).

Comment 5: Page 4, Regarding the quantification of lipids and nervonic acid, the authors described “FAMES were identified and quantified using commercial FAME standards purchased from Sigma-Aldrich (Shanghai, China)” . Were the FAMES in the samples calculated by the internal standard (specific chain length of FAME) or by the mixture of FAME standard mixture? Was the standard of nervonic acid used for quantifying nervonic acid in the samples, too?

Response: In this manuscript, FAMES were identified using commercial FAME standards and area normalization method ¹ was adopted to quantify nervonic acid and other FAMES (Visentainer JV. Analytical aspects of the flame ionization detector response of fatty acid esters in biodiesels and foods. *Quím Nova*. 2012, 35:274–279). Both internal standard and the standard of nervonic acid were used in our routine work, but not systematically used in this study.

Comment 6:

1) the result of expressing AtADS2 is described in Figure S3 not Figure 2E.

2) please suggest the potential reason for the opposite result of expressing AtADS2 in *S. cerevisiae* and *Y. lipolytica* with relevant references. The authors discussed the assumption with structural analysis of AtADS2 and mSCD which is not verified by experiment in this study.

3) The authors investigated the change of AtADS2' s conformation by site-specific mutation. How were these mutation sites decided?

Response:

1) This description was corrected to “Fig. S3”. (Page 25).

2) It may be the result of AtADS2 catalytic promiscuity

Previous studies aimed at determining enzyme activity using yeast (*S. cerevisiae*) expression gave evidence that AtADS2 was capable of catalyzing $\Delta 9$ or $\Delta 7$ desaturation of C16 and C18 saturated fatty acids, and it proves that enzyme functionality can be influenced by metabolic context. When the plastidial Arabidopsis 16:0 $\Delta 7$ desaturase FAD5 (AtADS3) was retargeted to the cytoplasm, regiospecificity shifted 70-fold, $\Delta 7$ to $\Delta 9$. Conversely, retargeting of two related cytoplasmic 16:0 $\Delta 9$ Arabidopsis desaturases (AtADS1 and AtADS2) to the plastid, shifted regiospecificity ~ 25 -fold, $\Delta 9$ to $\Delta 7$.¹ Through forward and reverse genetics, it was shown that AtADS2 is involved in the synthesis of the 24:1(n-9) and 26:1(n-9) components of seed lipids, sphingolipids, and the membrane phospholipids phosphatidylserine, and phosphatidylethanolamine, further research suggests that the 24-carbon and 26-carbon monounsaturated VLCFAs of Arabidopsis result primarily from VLCFA desaturation, rather than by elongation of long chain monounsaturated fatty acids.²

In the above studies, there is no analysis and research on the three-dimensional structure of AtADS2. We used the human stearoyl CoA desaturase (SCD) crystal structure (PDB ID: 4ZYO) as a template and obtained the three-dimensional structure of AtADS2 based on homologous modeling. The structural evaluation showed that the structure was highly reliable. Analysis found that its three-dimensional structure (including binding pocket and catalytic domain) is almost consistent with the typical C18:0-CoA desaturase SCD; Combining our experimental results in *Y. lipolytica*, from

the perspective of optimal catalytic substrates, we believe that AtADS2 should be attributed to SCD and not a specific C24:0-CoA desaturase. The optimal substrates for AtADS2 should be C18:0-CoA and C16:0-CoA.

Contrary to the traditional view that enzymes have specific catalytic functions, recent theoretical and experimental analyses have confirmed that enzyme catalytic promiscuity is a common natural evolutionary phenomenon.³⁻⁵ The specificity of enzymes is relative (i.e. relative specificity), which means that under specific catalytic conditions, the catalytic promiscuity of enzymes is inhibited, exhibiting specificity; As the external environment and/or its own sequence structure change, it triggers its catalytic promiscuity. *Y. lipolytica*, as an oil producing yeast, is associated with the metabolic context of *S. cerevisiae* is different; And compared to C24:0-CoA, there is a large amount of optimal substrate C18:0-CoA in *Y. Lipolytica* engineering strain, so AtADS2 is found in *Y. lipolytica* exhibits C18:0-CoA desaturation activity.

3) AtADS2 is almost identical to the typical C18:0-CoA desaturase SCD in terms of three-dimensional structure (Homologous modeling), and can achieve perfect matching when binding to C18:0-CoA. However, C24:0-CoA has longer hydrophobic hydrocarbon chains compared to C18:0-CoA, resulting in instability in its binding. So we hope to enhance the affinity of AtADS2 for C24:0-CoA by enhancing the hydrophobicity of its binding pocket periphery. Therefore, we chose to perform hydrophobic mutations on the polar amino acids binding pocket periphery in order to achieve n-9 desaturase activity towards C24:0-CoA.

Reference

1. Heilmann I, *et al.* Switching desaturase enzyme specificity by alternate subcellular targeting. *Proceedings of the National Academy of Sciences*, 2004, 101: 10266-10271.
2. Smith MA, Dauk M, Ramadan H, *et al.* Involvement of Arabidopsis ACYL-COENZYME A DESATURASE-LIKE2 (At2g31360) in the biosynthesis of the very-long-chain monounsaturated fatty acid components of membrane lipids. *Plant physiology*, 2013, 161(1): 81-96.

3. Baier F, Copp JN, Tokuriki N. Evolution of enzyme superfamilies: comprehensive exploration of sequence-function relationships. *Biochemistry*, 2016, 55(46): 6375-6388;
4. Hammer SC, *et al.* Design and evolution of enzymes for non-natural chemistry. *Curr Opin Green Sustain Chem*, 2017, 7:23-30.
5. Newton MS, Arcus VL, Gerth ML, *et al.* Enzyme evolution: innovation is easy, optimization is complicated. *Curr Opin Struct Biol*, 2018, 48:110-116.

Comment 7: Page 8, Figure 3 (c), please described the strain's abbreviation in the legend for better understanding of readers.

Response: We made changes accordingly (Page 7): “YL-3CgK, the CgKCS overexpressing strain; YL-3CgKE, expression of gELOVL6 in YL-3CgK; YL-3CgKEM, expressing an additional copy of each of gELOVL6 and MaOLE2 in YL-3CgKE.”.

Comment 8: Page 10. “Interestingly, the ratio of nervonic acid to lignoceric acid (C24:0) increased from 1.38 in YLNA6 to 2.23 in YLNA7 by disruption of the AMP-activated S/T protein kinase SNF1 (Fig. 5C and 5D),” YLNA7 was constructed by disrupting SNF1 and the overexpression of additional copy of CgKCS, therefore, the positive effect on nervonic acid production cannot be simply explained by SNF1 disruption. The comparison between YLNA7 and YLVL10 looks more reasonable, but still needs to be described the difference of random integration or homologous recombination during the strain construction.

Response: We agree that the reasons of the ratio of nervonic acid to lignoceric acid (C24:0) increased cannot be concluded according to current data. So, we only described the processes to cause the result in the revised manuscript (Page 9), “Interestingly, the ratio of nervonic acid to lignoceric acid (C24:0) increased from 1.38 in YLNA6 to 2.23 in YLNA7 by disruption of the AMP-activated S/T protein kinase *SNF1* and the overexpression of additional copy of CgKCS by random genomic integration”.

Comment 9: Page 11. Figure 5 (A), please indicate the fusion protein differently (eg. CgKCS-MaolE2).

Response: We used a short curve to link the fusion proteins (Figure 5A, Page10).

Comment 10: Page 13. Production of nervonic acid by two-stage fed-batch bioreactor. The authors optimized the media, mostly the C/N ratio for higher lipid accumulation. Please described why the C/N ratio is affecting the lipid accumulation in *Y. lipolytica* and how it was used for lipid production with relevant references.

Response: We described why the C/N ratio is affecting the lipid accumulation in *Y. lipolytica* and how it was used for lipid production with relevant references in the revised manuscript (Page 12).

Comment 11: Table S6 & S7, please add the C/N ratio in each condition.

Response: The C/N ratio was added in table S6 and S7.

Comment 12: Fig S8, Please compare the mass spectra of nervonic acid in standard and the samples.

Response: The mass spectra of nervonic acid in standard and the samples were compared in revised manuscript (Fig S8, Page 29).

Comment 13: Fig S9, for supporting the enlargement of ER after INO2 overexpression, can you please provide the approximate lengths with the standard deviation of ER from multiple images? Or multiple images from the same strain can show the enlargement of ER overall?

Response: We provided the approximate lengths with the standard deviation of ER from multiple images in revised manuscript (Fig S9, Page 30).

Minor typing error

Comment 1: Page 13. “The titer of nervonic acid increased 18.2% from 2.96 g/L to 3.5 g/L in YLNA9 (Fig. 7E).” -> increased by 18.2%

Comment 2: Page 17. Conclusion, “Enhancing the carbon flux toward fatty acid elongation efficiently...” -> toward

Response: The typing errors were corrected (page 12 and 16) in the revised manuscript.

REVIEWERS' COMMENTS:

Reviewer #1 (Remarks to the Author):

Since all the queries have been adressed by the authors, it can be accepted now.

Reviewer #2 (Remarks to the Author):

The authors modified the manuscript well.

Response to reviewers

Reviewer: 1

Comments to the Author

Nervonic acid, which benefits the treatment of neurological diseases and the health of brain, is a kind of functional fatty acid traditionally extracted from the plant. In the present study, the oleaginous yeast *Yarrowia lipolytica* was engineered for high-level production of nervonic acid by systematic metabolic engineering. The final engineered strain produced 17.3 g/L nervonic acid, the highest reported titer to date for de novo nervonic acid production. The methods are clear and the results are also well organized. For these reasons, I recommend to accept it if the following questions are addressed by the authors.

Comment 1: In Abstract section, the object after "expression of" and "disruption of" should be a gene rather than an enzyme.

Response: We changed the description to “expression of the genes encoding.....” and “disruption of the AMP-activated S/T protein kinase gene *SNFI*”. (please see Page 1, Abstract).

Comment 2: In Abstract section, the "Cg" of the "CgKCS" should be italicized. Other similar genes or enzymes should also be changed.

Response: The "Cg" of the "CgKCS" was italicized throughout the manuscript.

Comment 3: In Abstract section, the proportion of fatty acid is as important as the titer, especially for the separation and purification. Please give the proportion of nervonic acid in total fatty acids.

Response: The proportion of nervonic acid in total fatty acids (17.9% of TFA) was added (please see Page 1, Abstract).

Comment 4: Please provide the source of CgKCS as early as possible.

Response: The source *Cardamine graeca* of CgKCS was shown in “strains and plasmids” section of the Experimental chapter (Page 3).

Comment 5: The full name of "KCS" and "erucic acid" should be given when it first appears, and the abbreviations should be used instead when it appears later.

Response: According to the suggestion, the full name β -ketoacyl-CoA synthase of "KCS" and the abbreviation EA of "erucic acid" were used (Page 3). The corresponding abbreviations were used when they appear later (Page 5, 12, 15, and 16).

Comment 6: “for production of” should be “for the production of”.

Response: The expression was corrected (Page 1, Page 2, Page 7, Page 17, Page 18).

Comment 7: “The fatty acid profiles in triglycerides produced by *Y. lipolytica* mainly consist of palmitic acid (C16:0), palmitoleic acid (C16:1, ω -7), and oleic acid (C18:1, ω -9).” C18:2 is also one of the most important native fatty acids in *Y. lipolytica*.

Response: Linoleic acid (C18:2, ω -6) was added (Page 2).

Comment 8: In figure 1, please confirm if the substrate of Δ 12 desaturase is acyl-CoA or phospholipid.

Response: The substrate of Δ 12 desaturase is acyl-CoA (Refs: The structural basis of fatty acid elongation by the ELOVL elongases. *Nat Struct Mol Biol.* 2021, 28:512-520; Structural and mechanistic insights into the biosynthesis of unsaturated fatty acids. *IUBMB Life.* 2022, 74:1036-1051.).

Comment 9: Please confirm whether "triacylglycerol" or "triglyceride".

Response: The main difference between triacylglycerol and triglyceride is that triacylglycerol is the correct chemical name for an ester derived from glycerol bound to three fatty acids whereas triglyceride is the common name for the substance.

Triglycerides are the main constituent of the animal and vegetable fats in the diet. So, it is accurate to use triacylglycerol. We replaced "triglyceride" with "triacylglycerol" (Page 2).

Comment 10: Please confirm whether "of TFA" or "of the TFA".

Response: We used “of the TFA” instead of "of TFA". (Page 1, Page 13).

Comment 11: “To the best of our knowledge, the acylcoenzyme A desaturase-like protein AtADS2 in *A. thaliana* is the only reported enzyme able to catalyze C24:0-acyl-CoA to synthesize nervonic acid.” While the $\Delta 15$ desaturase from *Mortierella alpina* has also been reported to catalyze C24:0 to C24:1, as reported in Reference 71, therefore, the above description is suggested to be adjusted.

Response: The expression was revised to “To the best of our knowledge, the acyl-coenzyme A desaturase-like protein AtADS2 in *A. thaliana* and the $\Delta 15$ desaturase from *Mortierella alpina* were able to catalyze C24:0-acyl-CoA to synthesize nervonic acid (Page 7)”.

Comment 12: Fig. 4 and 5, the fatty acid profiles of Po1g-G3 strain should be added.

Response: The fatty acid profiles of Po1g-G3 strain were added in Fig. 4 and 5 in the revised manuscript.

Comment 13: The nervonic acid content of the strain YLNA7 should be presented.

Response: The nervonic acid content of the strain YLNA7 was presented in Fig.5C (Wathet blue column) and this result was described correspondingly in the revised manuscript. Please see the page 9 “The nervonic acid content in YLNA8 (17.3%) was slightly higher than YLNA7 (16.5%) (Fig. 5C and 5D).”

Comment 14: Why not combine some other strategies, like co-expressing INO2 and MoGPAT-88? This may further increase nervonic acid production.

Response: Thanks for this valuable suggestion and we will test it.

Comment 15: The T-test should be carried out. After all, some data are not significantly improved.

Response: The T-tests of data used in Fig.4 and Fig.5 were carried out (Page8 and Page10).

Comment 16: “Notably, a nervonic acid titer of 13.5 g/L in 5-L bioreactor⁷¹ was reported closely after this study posted on bioRxiv.” This description must be revised, as the Reference 71 was received on 20 March 2023, and Accepted on 1 April 2023, while the present study posted on bioRxiv was on 30 March 2023, it is after the Reference 71. Therefore, the description should be revised to be “Notably, a nervonic acid titer of 13.5 g/L in 5-L bioreactor⁷¹ was reported closely before this study posted on bioRxiv.” Or just do not compare the order, I suggest.

Response: We accepted this suggestion and using the following description “Notably, a nervonic acid titer of 13.5 g/L in 5-L bioreactor⁷¹ was reported closely before this study posted on bioRxiv.”

Comment 17: The authors mentioned that they have conducted purification and separation of nervonic acid, and generated a purity of 98.7%, however, the detailed steps and procedures was not mentioned. Therefore, I suggest to delete this description to avoid of misunderstanding. Also, the authors should clearly understand that the nervonic acid is accumulated in the form of lipid (triacylglycerol), so the final product form is the oil product, just as the plant extract oil form product. The nervonic acid percent of the total fatty acid or the total oil is a qualitative index of the final product. Therefore, in addition to consider the final g/L nervonic acid, we should consider another important product parameter of “% of the total fatty acid or the total oil”, please add this information of the final product.

Response: The aim of analyzing nervonic acid in free fatty acid form in this study was

to verify that nervonic acid can be separated from lignoceric acid (C24:0). The detailed methods and procedures were described in Experimental section. (From page 4 to 5). Additionally, the physiological and pharmacological functions of fatty acids are not necessarily in the form of TAG. Free fatty acids or ethyl ester forms of fatty acids are also important, such as docosahexaenoic acid ethyl ester (DHA-EE) and ethyl ester of nervonic acid. Several publications involving the functions of nervonic acid in free fatty acid or ethyl ester were summarized as follows:

- a) Nervonic acid ameliorates motor disorder with Parkinson's disease. In this study, free fatty acid NA (90%) was purchased from Hengke biotechnology Co. (Shanghai, China, CAS. no: 506-37-6). (Hu, D., Cui, Y. & Zhang, J. Nervonic acid ameliorates motor disorder in mice with Parkinson's disease. *Neurochem. J.* 2021, 15, 317-324)
- b) The cell viability was significantly increased in NA-treated cells, suggesting NA had a potential activity to upregulate the ROS elimination pathway to against oxidative stress. In this study, free fatty acid NA (>99%) was obtained from Nu-Chek Prep (MN, USA). (Umemoto H, Yasugi S, Tsuda S, Yoda M, Ishiguro T, Kaba N, Itoh T. Protective effect of nervonic acid against 6-hydroxydopamine-induced oxidative stress in PC-12 cells. *J Oleo Sci.* 2021, 70:95-102)
- c) Nervonic acid limits weight gain in a mouse model of diet-induced obesity. Increasing dietary NA may be an effective way to improve the management of obesity and associated metabolic complications including diabetes. In this study, ethyl ester of nervonic acid (purity greater than 99%+) was obtained from Nu-Chek Prep (Elysian, MN). (Keppley LJW, Walker SJ, Gademsey AN, Smith JP, Keller SR, Kester M, Fox TE. Nervonic acid limits weight gain in a mouse model of diet-induced obesity. *FASEB J.* 2020, 34:15314-15326)
- d) NA can also function as a non-competitive inhibitor of human immunodeficiency virus type-1 reverse transcriptase (HIV-1 RT) in a dose-dependent manner. In this study, nervonic acid (NA, analytical grade) were purchased from Wako (Kasai, N., Mizushina, Y., Sugawara, F., and Sakaguchi, K. Three-dimensional structural model analysis of the binding site of an inhibitor, nervonic acid, of both DNA

polymerase beta and HIV-1 reverse transcriptase. *J. Biochem.* 2922, 132, 819-828)

- e) NA can significantly ameliorate LPS-induced neuroinflammation and deterioration of learning and memory, and exerts a neuroprotective function through regulation of multiple gene transcription and metabolism pathways. In this study, NA ($\geq 99.0\%$ (GC)) were purchased from Sigma-Aldrich Co., St. Louis, MO, United States. (Wang X, Li Z, Li X, Liu X, Ying M, Cao F, Zhu X, Zhang J. Integrated metabolomics and transcriptomics reveal the neuroprotective effect of nervonic acid on LPS-induced AD model mice. *Biochem Pharmacol.* 2023, 209:115411.)

“The titer of nervonic acid and VLCFAs in YLNA9 reached 17.3 g/L (17.9% of the TFA) and 28.2 g/L (29.2% of the TFA), respectively (Fig. 9C), as far as we known, both of which were the highest reported levels.”. (page 12).

Reviewer: 2

Comments to the Author

Hang Su and colleagues engineered the oleaginous yeast *Yarrowia lipolytica* by rational metabolic engineering to improve the production of very long-chain unsaturated fatty acids, nervonic acid. Diverse strategies, introducing key enzymes (heterologous β -ketoacyl-CoA synthase, elongate, desaturase), enlarging a compartment for increasing lipid synthesis, and optimizing fermentation conditions successfully increased the production of nervonic acid with a high titer. This study shows several aspects which are necessary for producing specific lipids by metabolic engineering and several analyses including structural analysis of key enzymes that support the author's strategy well. The manuscript needs to be a bit more improved regarding the presentation of results depending on the strategies rather than showing all results without highlighting the conclusion of each strategy (eg. the effect of GPAT: ratio of LCFA/VLCFA, the effect of AtADS2: ratio of C18:1/C24:1).

Comment 1: In general, it would be great if the authors can add a small table or figure (either in each figure or in another new figure) to show the effect of each engineering step (ratio of nervonic acid in the TFA, titer of nervonic acid, ratio of LCFA/VLCFA as it is mentioned in discussion and conclusion).

Response: According to the suggestion, a summary table was added in the revised manuscript (Table 1, page 16).

Comment 2: The authors used specific gene loci like TGL4, GSY1, and PEX10 for homologous recombination to integrate heterologous genes. Some of the genes are highly involved in lipid metabolism in *Y. lipolytica*. So, the effect of disruption of them cannot be neglected in the interpretation of the result. Please add the description of this effect and modify the data interpretation accordingly.

Response: We have added the description the effect of disruption of specific gene loci in the revised manuscript (page 9).

Comment 3: Page 2, Figure 1. The authors described the heterologous enzyme in blue in the pathway. As some native enzymes of *Y. lipolytica* are also involved in some steps in the pathway, it would be better to add the specific enzyme or gene name (eg. CgKCS, MoGPAT).

Response: The specific enzymes or gene names were added according to the suggestion (Figure 1, page 2)

Comment 4: Page 3, The information about gene sequence is not fully described either in the main manuscript or supplementary manuscript (eg. CgKCS, AtFAE1, BtFAE1, etc). Please indicate the gene number in the database or amino acid sequence as supplementary information.

Response: We have added the gene sequences in supplementary data (Table. S8, Supplementary data Page 15-22).

Comment 5: Page 4, Regarding the quantification of lipids and nervonic acid, the authors described “FAMES were identified and quantified using commercial FAME standards purchased from Sigma-Aldrich (Shanghai, China)” . Were the FAMES in the samples calculated by the internal standard (specific chain length of FAME) or by the mixture of FAME standard mixture? Was the standard of nervonic acid used for quantifying nervonic acid in the samples, too?

Response: In this manuscript, FAMES were identified using commercial FAME standards and area normalization method ¹ was adopted to quantify nervonic acid and other FAMES (Visentainer JV. Analytical aspects of the flame ionization detector response of fatty acid esters in biodiesels and foods. *Quím Nova*. 2012, 35:274–279). Both internal standard and the standard of nervonic acid were used in our routine work, but not systematically used in this study.

Comment 6:

1) the result of expressing AtADS2 is described in Figure S3 not Figure 2E.

2) please suggest the potential reason for the opposite result of expressing AtADS2 in *S. cerevisiae* and *Y. lipolytica* with relevant references. The authors discussed the assumption with structural analysis of AtADS2 and mSCD which is not verified by experiment in this study.

3) The authors investigated the change of AtADS2' s conformation by site-specific mutation. How were these mutation sites decided?

Response:

1) This description was corrected to “Fig. S3”. (Page 25).

2) It may be the result of AtADS2 catalytic promiscuity

Previous studies aimed at determining enzyme activity using yeast (*S. cerevisiae*) expression gave evidence that AtADS2 was capable of catalyzing $\Delta 9$ or $\Delta 7$ desaturation of C16 and C18 saturated fatty acids, and it proves that enzyme functionality can be influenced by metabolic context. When the plastidial Arabidopsis 16:0 $\Delta 7$ desaturase FAD5 (AtADS3) was retargeted to the cytoplasm, regiospecificity shifted 70-fold, $\Delta 7$ to $\Delta 9$. Conversely, retargeting of two related cytoplasmic 16:0 $\Delta 9$ Arabidopsis desaturases (AtADS1 and AtADS2) to the plastid, shifted regiospecificity ~ 25 -fold, $\Delta 9$ to $\Delta 7$.¹ Through forward and reverse genetics, it was shown that AtADS2 is involved in the synthesis of the 24:1(n-9) and 26:1(n-9) components of seed lipids, sphingolipids, and the membrane phospholipids phosphatidylserine, and phosphatidylethanolamine, further research suggests that the 24-carbon and 26-carbon monounsaturated VLCFAs of Arabidopsis result primarily from VLCFA desaturation, rather than by elongation of long chain monounsaturated fatty acids.²

In the above studies, there is no analysis and research on the three-dimensional structure of AtADS2. We used the human stearoyl CoA desaturase (SCD) crystal structure (PDB ID: 4ZYO) as a template and obtained the three-dimensional structure of AtADS2 based on homologous modeling. The structural evaluation showed that the structure was highly reliable. Analysis found that its three-dimensional structure (including binding pocket and catalytic domain) is almost consistent with the typical C18:0-CoA desaturase SCD; Combining our experimental results in *Y. lipolytica*, from

the perspective of optimal catalytic substrates, we believe that AtADS2 should be attributed to SCD and not a specific C24:0-CoA desaturase. The optimal substrates for AtADS2 should be C18:0-CoA and C16:0-CoA.

Contrary to the traditional view that enzymes have specific catalytic functions, recent theoretical and experimental analyses have confirmed that enzyme catalytic promiscuity is a common natural evolutionary phenomenon.³⁻⁵ The specificity of enzymes is relative (i.e. relative specificity), which means that under specific catalytic conditions, the catalytic promiscuity of enzymes is inhibited, exhibiting specificity; As the external environment and/or its own sequence structure change, it triggers its catalytic promiscuity. *Y. lipolytica*, as an oil producing yeast, is associated with the metabolic context of *S. cerevisiae* is different; And compared to C24:0-CoA, there is a large amount of optimal substrate C18:0-CoA in *Y. Lipolytica* engineering strain, so AtADS2 is found in *Y. lipolytica* exhibits C18:0-CoA desaturation activity.

3) AtADS2 is almost identical to the typical C18:0-CoA desaturase SCD in terms of three-dimensional structure (Homologous modeling), and can achieve perfect matching when binding to C18:0-CoA. However, C24:0-CoA has longer hydrophobic hydrocarbon chains compared to C18:0-CoA, resulting in instability in its binding. So we hope to enhance the affinity of AtADS2 for C24:0-CoA by enhancing the hydrophobicity of its binding pocket periphery. Therefore, we chose to perform hydrophobic mutations on the polar amino acids binding pocket periphery in order to achieve n-9 desaturase activity towards C24:0-CoA.

Reference

1. Heilmann I, *et al.* Switching desaturase enzyme specificity by alternate subcellular targeting. *Proceedings of the National Academy of Sciences*, 2004, 101: 10266-10271.
2. Smith MA, Dauk M, Ramadan H, *et al.* Involvement of Arabidopsis ACYL-COENZYME A DESATURASE-LIKE2 (At2g31360) in the biosynthesis of the very-long-chain monounsaturated fatty acid components of membrane lipids. *Plant physiology*, 2013, 161(1): 81-96.

3. Baier F, Copp JN, Tokuriki N. Evolution of enzyme superfamilies: comprehensive exploration of sequence-function relationships. *Biochemistry*, 2016, 55(46): 6375-6388;
4. Hammer SC, *et al.* Design and evolution of enzymes for non-natural chemistry. *Curr Opin Green Sustain Chem*, 2017, 7:23-30.
5. Newton MS, Arcus VL, Gerth ML, *et al.* Enzyme evolution: innovation is easy, optimization is complicated. *Curr Opin Struct Biol*, 2018, 48:110-116.

Comment 7: Page 8, Figure 3 (c), please described the strain's abbreviation in the legend for better understanding of readers.

Response: We made changes accordingly (Page 7): “YL-3CgK, the CgKCS overexpressing strain; YL-3CgKE, expression of gELOVL6 in YL-3CgK; YL-3CgKEM, expressing an additional copy of each of gELOVL6 and MaOLE2 in YL-3CgKE.”.

Comment 8: Page 10. “Interestingly, the ratio of nervonic acid to lignoceric acid (C24:0) increased from 1.38 in YLNA6 to 2.23 in YLNA7 by disruption of the AMP-activated S/T protein kinase SNF1 (Fig. 5C and 5D),” YLNA7 was constructed by disrupting SNF1 and the overexpression of additional copy of CgKCS, therefore, the positive effect on nervonic acid production cannot be simply explained by SNF1 disruption. The comparison between YLNA7 and YLVL10 looks more reasonable, but still needs to be described the difference of random integration or homologous recombination during the strain construction.

Response: We agree that the reasons of the ratio of nervonic acid to lignoceric acid (C24:0) increased cannot be concluded according to current data. So, we only described the processes to cause the result in the revised manuscript (Page 9), “Interestingly, the ratio of nervonic acid to lignoceric acid (C24:0) increased from 1.38 in YLNA6 to 2.23 in YLNA7 by disruption of the AMP-activated S/T protein kinase *SNF1* and the overexpression of additional copy of CgKCS by random genomic integration”.

Comment 9: Page 11. Figure 5 (A), please indicate the fusion protein differently (eg. CgKCS-MaolE2).

Response: We used a short curve to link the fusion proteins (Figure 5A, Page10).

Comment 10: Page 13. Production of nervonic acid by two-stage fed-batch bioreactor. The authors optimized the media, mostly the C/N ratio for higher lipid accumulation. Please described why the C/N ratio is affecting the lipid accumulation in *Y. lipolytica* and how it was used for lipid production with relevant references.

Response: We described why the C/N ratio is affecting the lipid accumulation in *Y. lipolytica* and how it was used for lipid production with relevant references in the revised manuscript (Page 12).

Comment 11: Table S6 & S7, please add the C/N ratio in each condition.

Response: The C/N ratio was added in table S6 and S7.

Comment 12: Fig S8, Please compare the mass spectra of nervonic acid in standard and the samples.

Response: The mass spectra of nervonic acid in standard and the samples were compared in revised manuscript (Fig S8, Page 29).

Comment 13: Fig S9, for supporting the enlargement of ER after INO2 overexpression, can you please provide the approximate lengths with the standard deviation of ER from multiple images? Or multiple images from the same strain can show the enlargement of ER overall?

Response: We provided the approximate lengths with the standard deviation of ER from multiple images in revised manuscript (Fig S9, Page 30).

Minor typing error

Comment 1: Page 13. “The titer of nervonic acid increased 18.2% from 2.96 g/L to 3.5 g/L in YLNA9 (Fig. 7E).” -> increased by 18.2%

Comment 2: Page 17. Conclusion, “Enhancing the carbon flux toward fatty acid elongation efficiently...” -> toward

Response: The typing errors were corrected (page 12 and 16) in the revised manuscript.